# AI co-pilot bronchoscope robot

Jingyu Zhang[1,2,4], Lilu Liu [1,2,4], Pingyu Xiang[1,2], Qin Fang[1,2], Xiuping Nie[1,2], Honghai Ma[3], Jian Hu[3], Rong Xiong [1,2] ✉, Yue Wang [1,2] ✉ & Haojian Lu [1,2] ✉

The unequal distribution of medical resources and scarcity of experienced practitioners confine access to bronchoscopy primarily to well-equipped hospitals in developed regions, contributing to the unavailability of bronchoscopic services in underdeveloped areas. Here, we present an artificial intelligence (AI) co-pilot bronchoscope robot that empowers novice doctors to conduct lung examinations as safely and adeptly as experienced colleagues. The system features a user-friendly, plug-and-play catheter, devised for robot-assisted steering, facilitating access to bronchi beyond the fifth generation in average adult patients. Drawing upon historical bronchoscopic videos and expert imitation, our AI–human shared control algorithm enables novice doctors to achieve safe steering in the lung, mitigating misoperations. Both in vitro and in vivo results underscore that our system equips novice doctors with the skills to perform lung examinations as expertly as seasoned practitioners. This study offers innovative strategies to address the pressing issue of medical resource disparities through AI assistance.

Lung diseases, such as lung cancer, chronic obstructive pulmonary disease and pneumonia, represent a significant global health burden, with millions of individuals affected yearly[1,2]. Early detection and intervention are crucial to mitigate the impact of these diseases, reduce morbidity and improve patient outcomes[3]. Bronchoscopy, a minimally invasive diagnostic and therapeutic procedure, has emerged as an essential tool in detecting, treating and managing various lung diseases[4–6]. During bronchoscopy, the bronchoscope should be inserted and manipulated gently to avoid abrupt or forceful movements, which can cause discomfort or injury to airway structures. Additionally, maintaining a central position during bronchoscopy allows better visualisation of the airway anatomy and helps prevent injury to the airway mucosa or other structures. This is especially important when navigating through tight or tortuous airways, strictures, or areas with masses. However, this procedure requires a high level of skill and experience, resulting in a significant disparity in the quality of care provided by expert and novice doctors[7,8]. The availability of bronchoscopic services is predominantly limited by the need for more experienced doctors in underdeveloped regions, resulting in a critical barrier to health care access for vulnerable populations[9]. The

need for specialised training and expertise in bronchoscopy further exacerbates this issue, as many health care systems in these areas need help to support the development and maintenance of such skills. Consequently, this knowledge gap hinders the establishment and expansion of bronchoscopic services in underprivileged settings, perpetuating the cycle of inadequate health care provision for lung disease patients[10,11].

Innovations combining the precision and dexterity of robotic systems with the guidance of expert doctors could help to resolve these clinical and technical challenges[12–14]. The development of robotic platforms and devices for bronchoscopy has seen significant progress in recent years, with systems such as the Monarch Platform[15] and the Ion Endoluminal System[16] leading the way. The Monarch Platform is equipped with an internal bronchoscope catheter with a 4.2 mm diameter and an external sheath of 6 mm. Its subtle steering control and flexibility, allowing deeper access into the peripheral regions of the lungs, surpass those of conventional bronchoscopes[17] (9th vs. 6th airway generations). The Ion Endoluminal System boasts a fully articulated 3.5 mm catheter with a 2 mm working channel, enhanced stability, superior flexibility and the added advantage of shape

[1]State Key Laboratory of Industrial Control and Technology, Zhejiang University, 310027 Hangzhou, China. [2]Institute of Cyber-Systems and Control, Department of Control Science and Engineering, Zhejiang University, 310027 Hangzhou, China. [3]Department of Thoracic Surgery, First Affiliated Hospital, School of Medicine, Zhejiang University, 310009 Hangzhou, China. [4]These authors contributed equally: Jingyu Zhang, Lilu Liu. ✉e-mail: rxiong@zju.edu.cn; ywang24@zju.edu.cn; luhaojian@zju.edu.cn

perception[18]. Notably, studies indicate that these platforms exhibit a favourable diagnostic yield, ranging from 81.7% to 92%, for lung nodules with sizes between 14.8 mm and 21.9 mm. Moreover, the complication rates reported are minimal[19–21]. These findings suggest that these platforms can play a transformative role in the future management of pulmonary conditions. In addition to the Monarch and Ion platforms, several other bronchoscope robotic systems are under development by academic institutions or have entered early-stage research to address sensing and control issues for doctors[22–27]. Nevertheless, despite its advantages, current telerobotic bronchoscopy faces several challenges, including a steep learning curve and lack of autonomy.

At present the integration of artificial intelligence (AI) techniques into bronchoscopy is further expanding the horizons of this burgeoning field[28]. By leveraging advanced algorithms, such as machine learning and computer vision technologies[29], researchers are developing image-guided navigation systems that process and interpret bronchoscopic imagery[30], facilitating real-time localisation[31], tracking[32] and interventional path planning[33] for endoscopy and enabling precise navigation within the bronchial tree. These software systems enhance the accuracy and efficiency of bronchoscopic procedures. Furthermore, by providing automated, continuous guidance throughout such a procedure[34], an image-guided system can help reduce the cognitive load on the operating doctor, allowing the doctor to focus on other critical aspects of the procedure[35]. However, these systems present safety concerns during bronchoscopic procedures because they rely on bronchoscope localisation in preoperative CT[36–38], which may result in misregistration and unsafe steering of the robot due to the limited field of view and body–CT visual discrepancies. Concerns have been raised about the risk of complications, such as pneumothorax and bleeding, underlining the need for ongoing research on and optimisation of these platforms.

We report an AI co-pilot bronchoscope robot for safe pulmonary steering. At the hardware level, we have designed a bronchoscope robot with a quick catheter replacement feature (utilising thin catheters for deep lung examination and thick catheters for examination and biopsy) based on magnetic adsorption, offering advanced performance that meets clinical requirements. At the software level, we have developed an AI–human shared control algorithm based on learning from experts, which can accommodate discrete human interventions while reducing the reliance on doctor expertise. Overall, the presented robotic system enhances safety while maintaining efficiency in bronchoscopy by providing novice doctors with increased autonomy. Through tests of this bronchoscope system on realistic human bronchial phantoms with simulated respiratory behaviour performed by a novice doctor with the AI co-pilot and by an expert, we demonstrate that our system enables novice doctors to access different bronchi proficiently. Subsequently, to validate the safety and effectiveness of our system under physiologically relevant conditions, we assess the system's steering capability in vivo using a live porcine lung to mimic the human bronchus. Our system carries the potential to improve the diagnosis and management of pulmonary disorders. It is anticipated that the cost and logistical barriers associated with the adoption of such platforms will decrease in the future, helping to overcome the challenge of medical resource disparities and contributing to the improvement of global health outcomes.

## Results

### Design of the AI co-pilot bronchoscope robot

Figure 1 provides an overview of our AI co-pilot bronchoscope robot (Supplementary Notes 1–3 and Supplementary Figs. 1–4), which is designed to be deployed in clinical settings for bronchoscopic procedures (Supplementary Movie 1). The bronchoscope robot is integrated with a robotic arm and placed alongside the operating table, and it is teleoperated by a doctor using a remote-control console to steer the bronchoscopic catheter. The robotic arm is used to adjust the intubation posture of the catheter for lung steering. The steering control system is composed of four linear motors for tendon actuation and four force sensors to measure the actuation force. To facilitate user-friendly replacement of the catheter, the steering control system and catheter are designed to connect by magnetic adsorption. The bronchoscopic catheter is composed of a proximal section with high stiffness and a distal section with low stiffness. A braided mesh is used in the proximal section for increased stiffness, while the distal section is composed of many small hinge joints for steering control. Both sections are covered with a thin thermoplastic urethane (TPU) layer for waterproofing. The tip of the bronchoscopic catheter consists of two LED lights for lighting and a microcamera for observation. Two types of catheters are designed, with diameters of 3.3 mm (with a 1.2 mm working channel) and 2.1 mm (without a working channel), both of which are smaller than those of the Monarch Platform and the Ion Endoluminal System, enabling access to bronchi of the ninth generation or deeper in average adult patients.

To improve the quality and consistency of bronchoscopic procedures and enable novice doctors to perform bronchoscopy as safely and proficiently as experienced specialists, we have designed an AI–human shared control algorithm to minimise the risk of damaging surrounding tissues while maintaining efficiency. Figure 2a illustrates an overview of the algorithm workflow. The algorithm's core is a policy network that takes a bronchoscopic image and a discrete human command (up, down, left, right, or forward) as input to predict a steering action (pitch and yaw angle rates) for the robot's orientation, which can be converted into tendon actuation through inverse kinematics and a low-level controller. As shown in Fig. 2b, the policy network training process consists of three steps: (a) establishment of a virtual bronchoscopy environment, (b) data preparation, and (c) Sim2Real adaptation. In the first step, an airway model is segmented from the preoperative CT volume to establish a virtual bronchoscopy environment. The airway centrelines are extracted by means of the Vascular Modeling Toolkit (VMTK) to serve as reference paths. By simulating the bronchoscope robot in this virtual environment, we can render its observed image and depth. Supplementary Note 4 presents the virtual environment and simulated robot configurations. In the second step, human commands and actions for each image are automatically generated for model supervision by an artificial expert agent (AEA) guided by privileged robot pose information and the reference airway centrelines, resulting in training samples consisting of images, depths, human commands and steering actions. For the third step, we propose a Sim2Real adaptation module to enhance the diversity and photorealism of the training samples. The domain adaptation part of this module translates rendered images into a more realistic style while preserving the bronchial structure by means of depth supervision, ensuring that the corresponding action supervision remains invariant. The domain randomisation part randomly alters the image appearances or adds noise to the human commands. Based on the dataset prepared as described above, a data aggregation algorithm (DAgger)[39] is employed for on-policy artificial expert imitation to eliminate distribution mismatch. Because every training sample is generated automatically, the entire training process is intervention-free. In practice, the input rendered images and AEA-annotated commands are replaced with real images and a novice doctor's commands in the real bronchoscope robot, driving the policy network to navigate the robot through the airway safely and smoothly. The discrete human commands are mapped to five regions of the teleoperator (Supplementary Fig. 3b), reducing the level of human intervention and the cognitive load compared to conventional teleoperated robots with continuous human guidance.

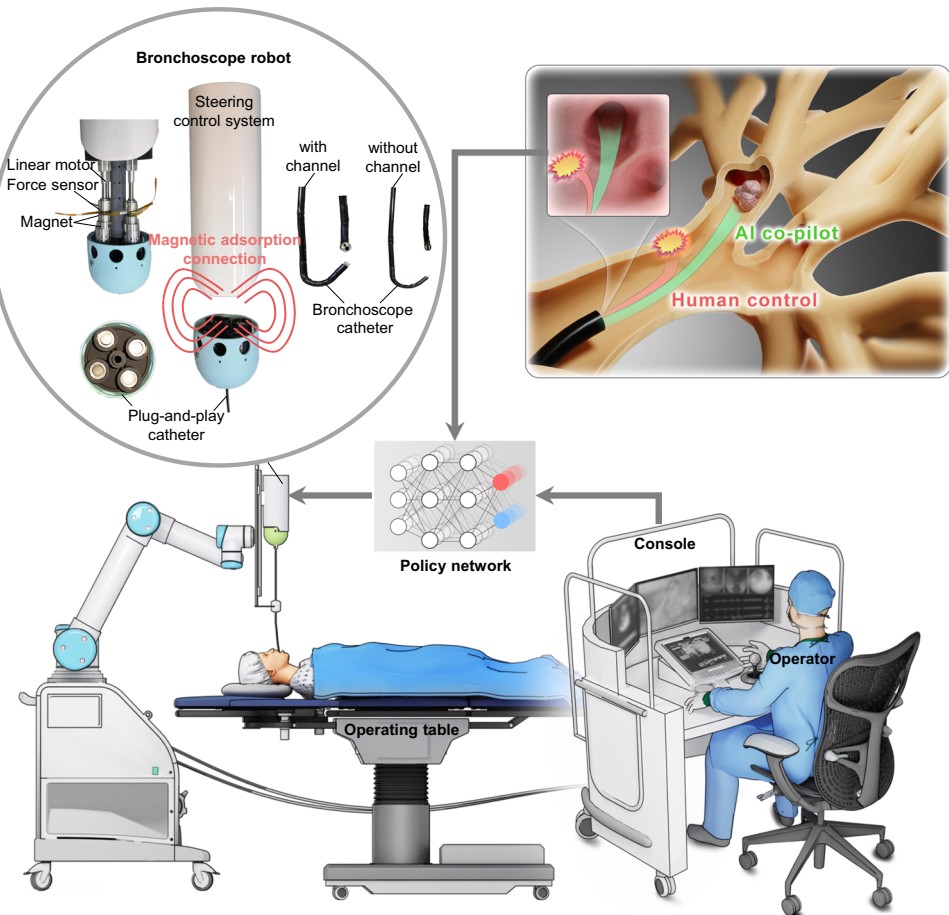

**Fig. 1 | Overview of our AI co-pilot bronchoscope robot deployed in a clinical setting for bronchoscopic procedures.** The bronchoscope robot is integrated with a robotic arm and placed alongside the operating table, and it is teleoperated by a doctor at a remote-control console to steer the bronchoscopy catheter for lung examination. Two types of catheters are designed, with diameters of 3.3 mm (with a 1.2 mm working channel) and 2.1 mm (without a working channel), enabling access to bronchi of the fifth generation or deeper in average adult patients. An AI–human shared control policy is designed to minimise the risk of damage to surrounding tissues and improve the overall safety profile.

## Simulation results and in vitro evaluation

We quantitatively assessed the performance of the proposed method through simulation experiments, in which airway models containing up to 5th-generation bronchi were utilised for training and evaluation (Fig. 3a). Two kinds of bronchoscopic images were rendered from airway models with pink and realistic textures, referred to as Sim-style and Real-style images, respectively (Fig. 3b). The realistic textures were generated by extracting actual bronchial wall textures from real historical clinical bronchoscopic videos. The training environments established based on the two airway models contained 74 and 84 reference paths, respectively. The test environment was built on the basis of another airway model with a realistic texture that contained 60 paths (Supplementary Fig. 5). The policy network trained using Sim-style images with domain adaptation and randomisation (Sim+A + R) achieved the highest success rate (calculated as the ratio of the numbers of successful paths to all paths, detailed definitions of which can be found in Supplementary Note 9 and Supplementary Fig. 17) of ~93.3% on the test paths (Fig. 3c). This performance surpassed that of networks trained without domain randomisation (Sim+A; ~80.0%), without any domain adaptation or randomisation (Sim; ~31.8%), using only Real-style images (Real; ~81.8%), and employing the baseline domain adaptation approach[40] (Sim+A(b); ~71.8%). Similarly, our Sim +A + R method exhibited the highest successful path ratio (calculated as the ratio of the completed path length over the total path length) of ~98.9 ± 4.7%, surpassing those of Real (~96.5 ± 8.0%), Sim+A (~96.4 ± 7.4%), Sim+A(b) (~92.9 ± 12.4%) and Sim (~75.2 ± 22.6%) (Fig. 3d). In terms of the trajectory error−the Euclidean distance between the predicted and reference paths−Sim+A + R also demonstrated the lowest error of ~1.04 ± 0.21 mm, compared to ~1.23 ± 0.28 mm for Real, ~1.37 ± 0.26 mm for Sim+A, ~2.57 ± 0.54 mm for Sim+A(b), and ~3.36 ± 0.66 mm for Sim (Fig. 3e). Specific results on each path are shown in Supplementary Fig. 6b. Notably, the Sim method underperformed, indicating overfitting of the policy network in the Sim-style image domain. Sim+A and Real showed similar performances, emphasising the effectiveness of our domain adaptation module. Sim+A outperformed Sim+A(b), validating the advantage of preserving the bronchial structure during domain adaptation. Sim +A + R achieved the best results, demonstrating that our proposed domain adaptation and randomisation procedures enable successful knowledge transfer between simulation and reality. The generalisation ability of Sim+A + R has also been tested on airway models from 10 different patients (Supplementary Fig. 7).

The domain adaptation module is necessary for translating Sim-style images (the source domain) into realistic images (the target domain). Thus, we qualitatively and quantitatively evaluated the image translation performance of our method. We selected three types of unpaired realistic images to represent target domains for training and evaluation, namely Real-, Phantom- and Clinical-style images (Supplementary Note 5 and Supplementary Fig. 8a). These images were collected from virtual airway models rendered with realistic texture

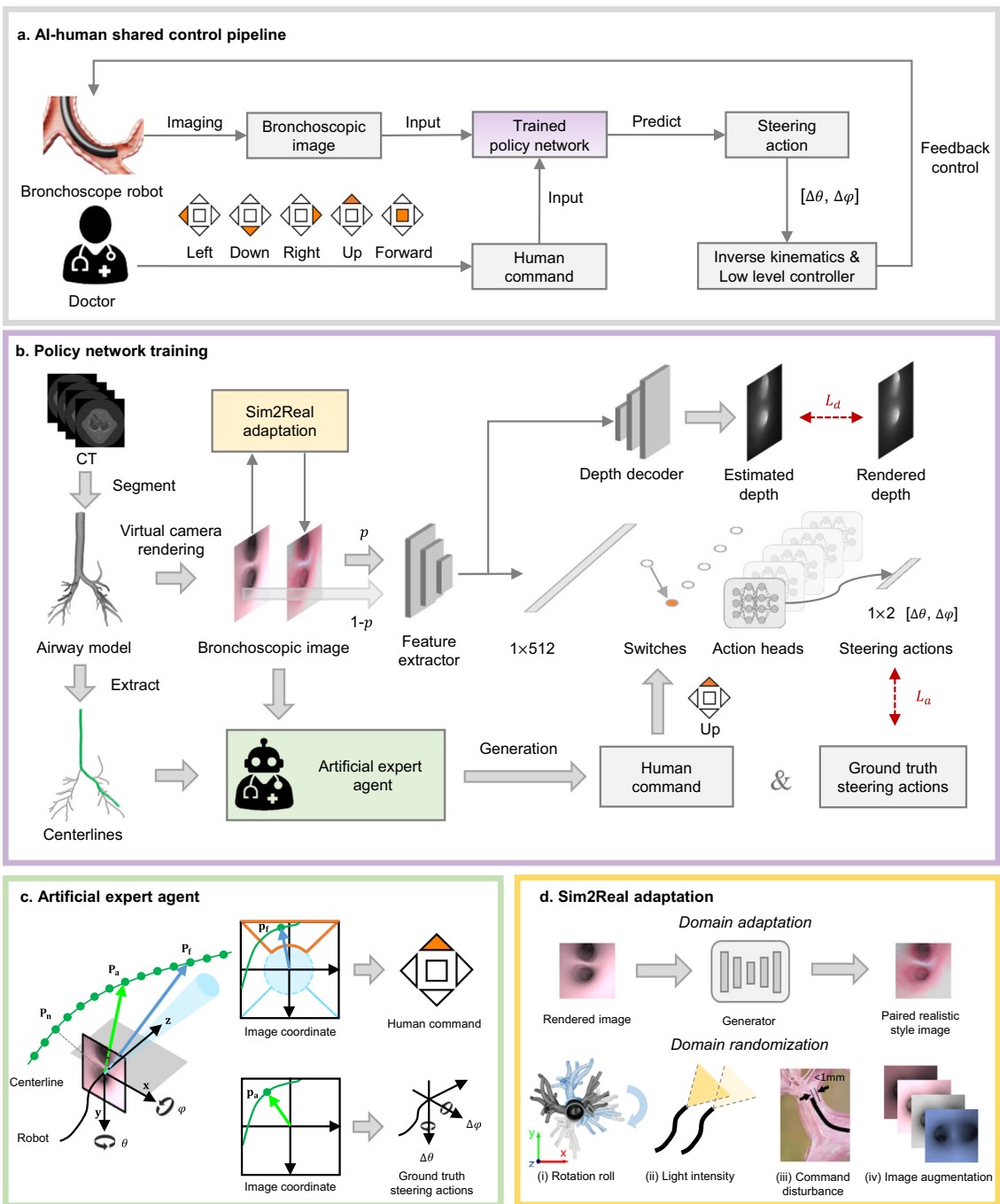

**Fig. 2 | AI–human shared control algorithm and training strategy. a** Overview of the AI co-pilot working pipeline. The bronchoscope robot acquires images during the procedure, and the doctor gives discrete human commands to determine the robot's direction. Both the images and human commands are input into a trained policy network to predict steering actions to control the catheter tip to remain centred at the bronchial lumen for safety, forming a closed-loop control system with inverse kinematics and a low-level controller integrated. (The "doctor icon" is used with permission from the AdobeStock Standard License.) **b** Policy network and training strategy. The policy network features a multi-task structure, where the main task is steering action prediction and the side task is depth estimation. To train the network, first, an airway model is segmented from the CT to establish a virtual environment. Airway centrelines are extracted as reference paths. Bronchoscopic images and depths are observed by a simulated robot through rendering. Second, an artificial expert agent automatically generates human commands and actions for supervision. Third, a Sim2Real adaptation module enhances the diversity and photorealism of the training images. **c** Artificial expert agent. This agent has priority to access all necessary information during the training process. Waypoints $P_a$ and $P_f$ are decision points. $P_a$ is used to determine the ground-truth steering action by calculating rotation angles from the robot's current posture. $P_f$ is located farther away than $P_a$ and is used to determine the human command by projecting this point into the image coordinate system of the robot, which is divided into five regions representing five discrete human commands. **d** Sim2Real adaptation module. This module first translates the rendered images into a realistic style while preserving the bronchial structure. Then, the generalisation ability of the policy network is enhanced by: (i) random rotation of the airway model's roll angle, (ii) random adjustment of the bronchoscope's light intensity, (iii) the addition of random noise to the human commands when the distance between the robot and the bronchial wall is <1 mm, and (iv) random alteration of the input images' attributes.

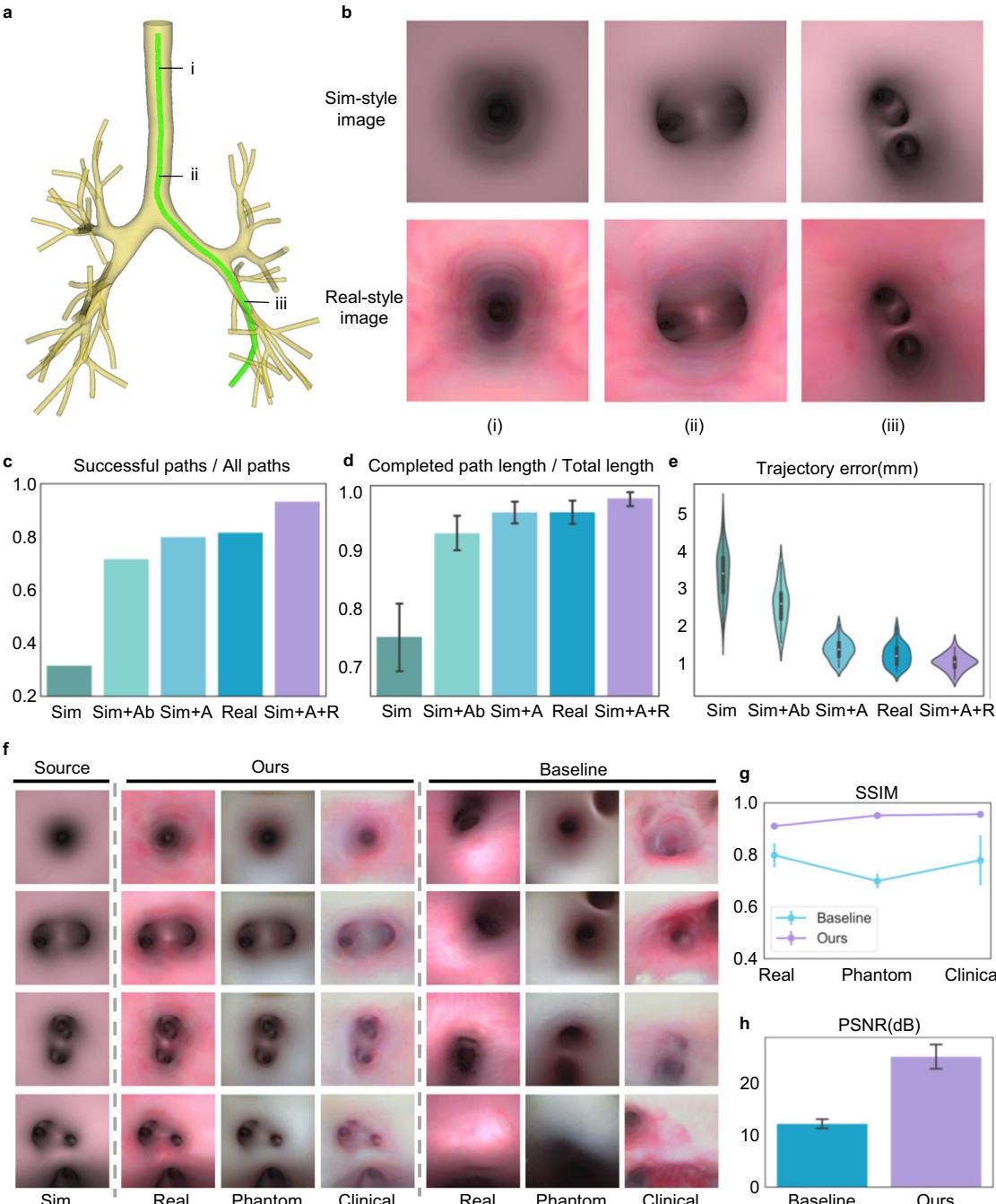

**Fig. 3 | Results of simulation experiments. a** An example of an airway model containing 5th-generation bronchi with a reference path for bronchoscopy. **b** Two styles of images rendered from airway models with different texture mapping. Sim-style images have a pink texture, while real-style images contain realistic textures. Three positions are depicted along the reference path: (i) the entrance of the trachea, (ii) the intersection of the trachea and the main bronchus, and (iii) the lower bronchus. **c** Success rates (i.e., the ratio of the numbers of successful paths to all paths, reflecting the generalisation ability of each method for reaching different branches of the bronchial tree). **d** Successful path ratios (i.e., the completed path length over the total path length of every path, reflecting the coverage ability of each method for the whole bronchial tree). Data are presented as mean values ± 95% confidence interval (CI), and the number of independent paths $n = 60$. **e** Trajectory errors of different methods when run in the test environment with realistic textures, containing $n = 60$ independent paths for evaluation. White circles indicate median, edges are 25th and 75th percentiles, whiskers indicate range. **f** Qualitative image translation results of different methods, where Sim represents the Sim-style images corresponding to the source domain and Real, Phantom and Clinical denote three styles of realistic images serving as target domains. The training datasets for the source and target domains are unpaired. AttentionGAN is chosen as the baseline method for unpaired image translation. Detailed illustrations of the training datasets and image translation results are given in Supplementary Fig. 8. **g** Structural similarity index measure. Data are presented as mean values ± standard deviation (SD), the number of independent experiments $n = 227$. **h** Peak signal-to-noise ratio results of different methods, containing $n = 227$ independent experiments presented as mean values ± SD. Higher SSIM and PSNR values indicate better structure-preserving properties.

mapping, real bronchoscopy in silicone phantoms, and real bronchoscopy in live pigs, respectively. As shown in Fig. 3f and Supplementary Fig. 8b, our method successfully performs image translation without changing the bronchial structure. Our method outperforms AttentionGAN[32] in terms of the structural similarity index measure (SSIM) across Real-style (0.91 vs. 0.80), Phantom-style (0.95 vs. 0.70) and Clinical-style (0.96 vs. 0.78) translations, demonstrating the superior structure-preserving property of our method. Additionally, our approach achieves a higher peak signal-to-noise ratio (PSNR) than AttentionGAN (25.16 dB vs. 12.25 dB) between translated and target images in the Real style (Fig. 3g). We conclude that our method can successfully generate images with the target domain style while preserving the bronchial structure, facilitating Sim2Real adaptation.

To assess the proposed AI co-pilot bronchoscope robot, experiments were conducted on a bronchial phantom made of silica gel replicating structured derived from human CT lung data (Fig. 4a). A crank-rocker mechanism-based breathing simulation system (Supplementary Note 7 and Supplementary Fig. 10) was designed to emulate human respiration (15 cycles per minute). Two bronchial phantoms with distinct bronchial structures were employed for in vitro evaluation (Fig. 4b). An expert (chief doctor) and a novice doctor (medical intern) were invited to perform bronchoscopic procedures using the robot without the AI co-pilot as a benchmark, while another novice doctor (attending doctor) also participated using the robot with the AI co-pilot. All procedures were performed on the same path using the teleoperator. The medical intern had no experience with bronchoscopy, while the attending doctor had a little experience (<5 years and <100 cases per year, compared to the chief doctor, who had >20 years of experience and >200 cases per year), as indicated in Supplementary Table 1. They were both presented with two demonstrations of robotic intubation, with and without the AI co-pilot, to learn how to operate the system. During the evaluation, the medical intern performed teleoperated bronchoscopy without the AI co-pilot along the bronchus path for three trials, with image errors decreasing from $62.39 \pm 1.91$ pixels to $43.54 \pm 2.01$ pixels (Fig. 4c, Supplementary Note 8 and Supplementary Figs. 11–13). Under the same conditions, the expert achieved more precise and stable bronchoscopic operation (with an image error of $31.45 \pm 1.19$ pixels). Assisted by our AI–human shared control algorithm, the attending doctor achieved even better bronchoscopic operation performance ($17.63 \pm 0.46$ pixels) than the expert in the first trial. Next, the expert and the attending doctor carried out experiments on both sides of two phantoms, with the attending doctor using the robot with the AI co-pilot (Supplementary Figs. 14–16 and Supplementary Movie 2 and 3). Detailed operation error comparisons are depicted in Fig. 4d, e. In Phantom 1, AI-assisted operation ($19.14 \pm 0.50$ pixels) exhibited a significantly lower operation error than expert operation ($38.84 \pm 0.84$ pixels). In Phantom 2, despite a small initial error, as shown in the Path 66 results, expert operation failed to maintain a low error during insertion into deeper bronchial airways, while AI-assisted operation consistently maintained a low image error, keeping the bronchoscope centred in the image. In addition, a specific analysis of AI control performance and mode switching between AI and teleoperation is described in Supplementary Notes 11 and 12 and Supplementary Figs. 20–23.

### In vivo demonstration with a live porcine lung model

We further evaluated the performance of the AI co-pilot bronchoscope robot in a minipig, whose bronchial structure closely resembles that of the human bronchus. The pig was purchased from Zhuhai BestTest Bio-Tech Co., Ltd., solely based on the pig's health condition (Supplementary Note 13). The Wuzhishan pig was female and three months old. The protocols for animal experiments were approved by the Institutional Animal Care and Ethics Committee of Zhuhai BestTest (IAC(S)2201003-1). During the experiment, the bronchoscope catheter was inserted through the oropharynx into the pig's bronchial airways,

while the doctor, seated at the console, controlled the robot to accomplish teleoperation and steering. Prior to clinical trials, a physical examination and a whole-body CT scan were conducted to reconstruct the bronchial structure (Fig. 5a) and ensure the pig's good health.

We selected two porcine bronchus paths for in vivo demonstration and conducted clinical trials with the expert and the attending doctor, the latter of whom was assisted by the AI co-pilot (Supplementary Fig. 24). As seen in the endoscopic images presented in Fig. 5b and Supplementary Movie 4, both doctors achieved steering through the porcine bronchus to beyond the 5th-generation bronchi (diameter ~2.5 mm), yielding almost identical visualisation results. As seen from the actuation displacement and actuation force (Fig. 5c) measurements during bronchoscopy, the attending doctor achieved smoother steering with the AI co-pilot than the expert did. To quantitatively analyse the control effects of the two volunteers, the mean values and fluctuation ranges of the actuation displacement and actuation force are illustrated in Fig. 5d. It is evident that AI-assisted steering resulted in a lower mean value and fluctuation range than the expert's operation overall. The operation error comparison further indicates that our proposed AI co-pilot bronchoscope robot maintains better bronchus centring than expert teleoperation. Based on an analysis of eight repeated trials on the live porcine lung, the attending doctor could perform bronchoscopy with a $11.38 \pm 0.16$ pixel operation error in collaboration with the AI co-pilot, achieving performance as good as or even better than that of the expert ($16.26 \pm 0.27$ pixels). Considering the pixel-to-millimetre calibration results reported in Supplementary Note 9 and Supplementary Fig. 18, the AI co-pilot group could achieve a mean 3D positioning error of less than 0.73 mm in all procedures. To further characterise the autonomy properties of the proposed system, we compared the number of doctor interventions between the attending doctor with the AI co-pilot and the expert. The statistical results in Fig. 5g demonstrate that the number of interventions, as defined in Supplementary Note 9, with the use of the AI co-pilot was significantly lower than that during expert teleoperation, greatly reducing the doctor's physical exertion and cognitive load during the bronchoscopic operation and further illustrating the autonomy of our proposed robotic bronchoscopy system. Furthermore, the NASA Task Load Index survey (Supplementary Note 10 and Supplementary Fig. 19) was also completed for a comprehensive assessment of human workload, demonstrating a significant reduction in both physical and mental burden when using our AI-assisted system.

## Discussion

Bronchoscopic intervention is preferred for sampling suspected pulmonary lesions owing to its lower complications. Recently, robot-assisted technologies, such as the Monarch Platform and the Ion Endoluminal System, have been introduced into bronchoscopic procedures to enhance manoeuvrability and stability during lesion sampling. However, due to the high cost of robotic bronchoscope systems and the expertise needed by doctors, the proliferation of this technology in underdeveloped regions is limited. Our study presents a low-cost comprehensive AI co-pilot bronchoscope robot (Supplementary Table 2) to improve the safety, accuracy, and efficiency of bronchoscopic procedures. The proposed system, which incorporates a shared control algorithm and state-of-the-art domain adaptation and randomisation approaches, bridges the gap between simulated and real environments, ensuring generalisability across various clinical settings. Moreover, this AI co-pilot bronchoscope robot enables novice doctors to perform bronchoscopy as competently and safely as experienced specialists, reducing the learning curve for bronchoscopic procedures and ensuring a consistent quality of care.

Our in vitro and in vivo evaluation results demonstrate the efficacy of the proposed AI co-pilot bronchoscope robot in achieving insertion into deep bronchial airways with high precision and reduced

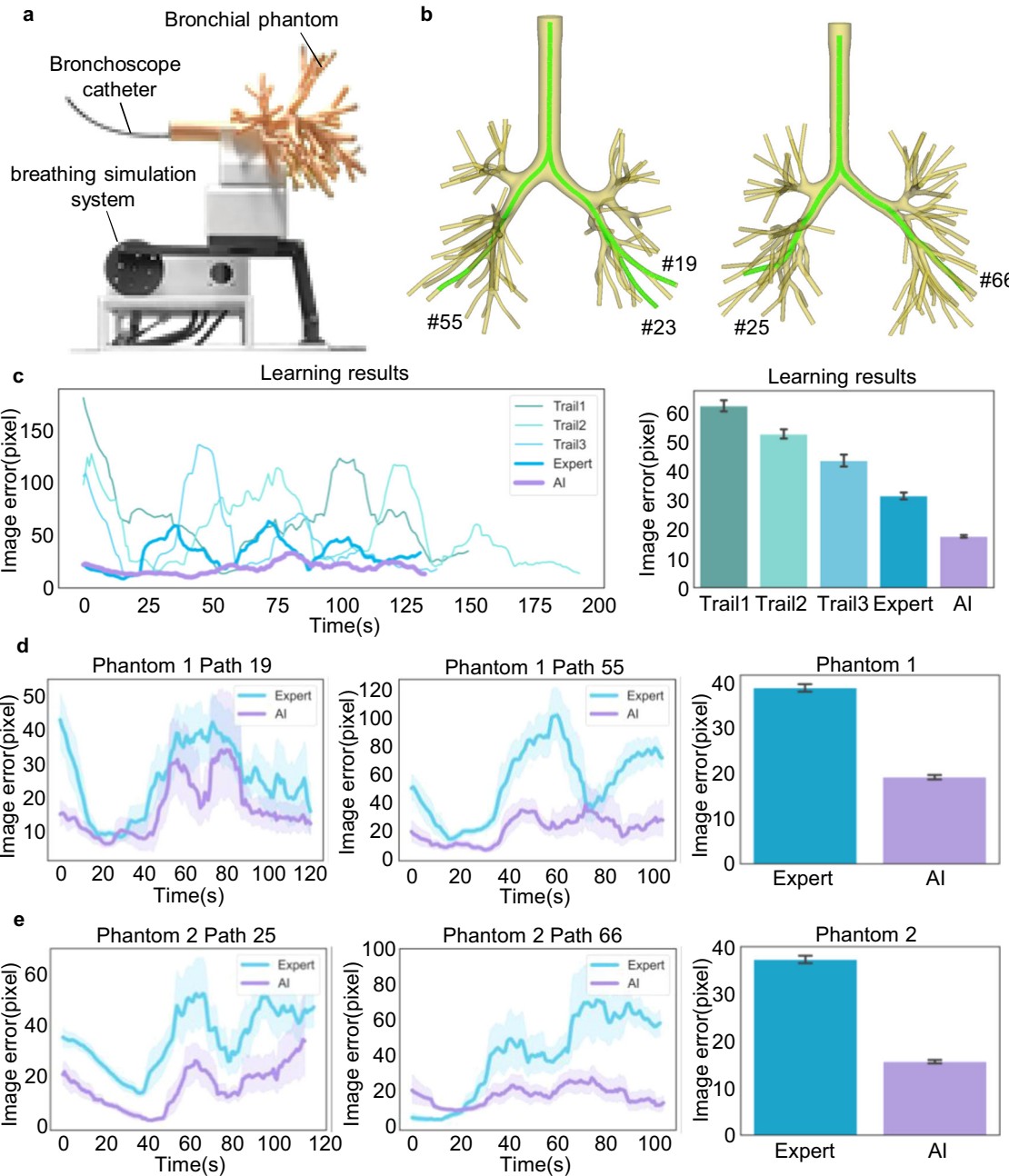

**Fig. 4 | Results of in vitro experiments. a** In vitro experimental scenario under breathing simulation, for which the detailed design and motion analysis are illustrated in Supplementary Fig. 10. **b** The airway models corresponding to the two bronchial phantoms. **c** Learning process of the novice doctors. Path 23 in Phantom 1 was selected for bronchoscopy. Two novice doctors (a medical intern and an attending doctor) were invited to learn teleoperated bronchoscopy with and without the AI co-pilot from demonstrations. The medical intern performed three bronchoscopy trials without the AI co-pilot. Then, the expert (without the AI co-pilot) and the attending doctor (with the AI co-pilot) separately performed bronchoscopy. Image errors were recorded during procedures. The numbers of recorded frames are $n = 3928, 5162, 3693, 3503$ and $3544$ for five independent experiments from Trial 1 to AI, respectively. Bar plots show mean ± 95% CI of image errors. **d** Comparison of results in Phantom 1. Paths 19 and 55 were selected for

bronchoscopy, covering both sides of Phantom 1. The expert without the AI co-pilot and the attending doctor with the AI co-pilot performed bronchoscopy separately. The numbers of recorded frames are $n = 6014$ for Expert and $n = 6094$ for AI. **e** Comparison of results in Phantom 2. Paths 25 and 66 were selected for bronchoscopy. The numbers of recorded frames are $n = 6180$ for Expert and $n = 5811$ for AI. The results reveal that the novice doctor with the AI co-pilot could achieve and maintain a smaller image error than the expert during the bronchoscopy procedure. Line plots with error bands depict time-varying mean image errors ± 95% CI within a time window of 20 frames during each procedure. Bar plots depict the mean image error and the 95% confidence interval for all frames during the two bronchoscopy procedures in each phantom, i.e., the mean ± 95% CI image error in Phantom 1, containing Path 19 and Path 55, and in Phantom 2, containing Path 25 and Path 66.

intervention from the operator. Notably, the AI–human shared control algorithm maintains better bronchus centring and exhibits lower operation errors than an expert operator, validating the robustness and clinical potential of our approach. Additionally, the domain adaptation and randomisation techniques effectively mitigate

overfitting and facilitate seamless knowledge transfer between the simulated and real image domains, ultimately contributing to the overall success of the system.

Despite these promising results, there are several areas for future research and development. Our bronchoscope robot relies

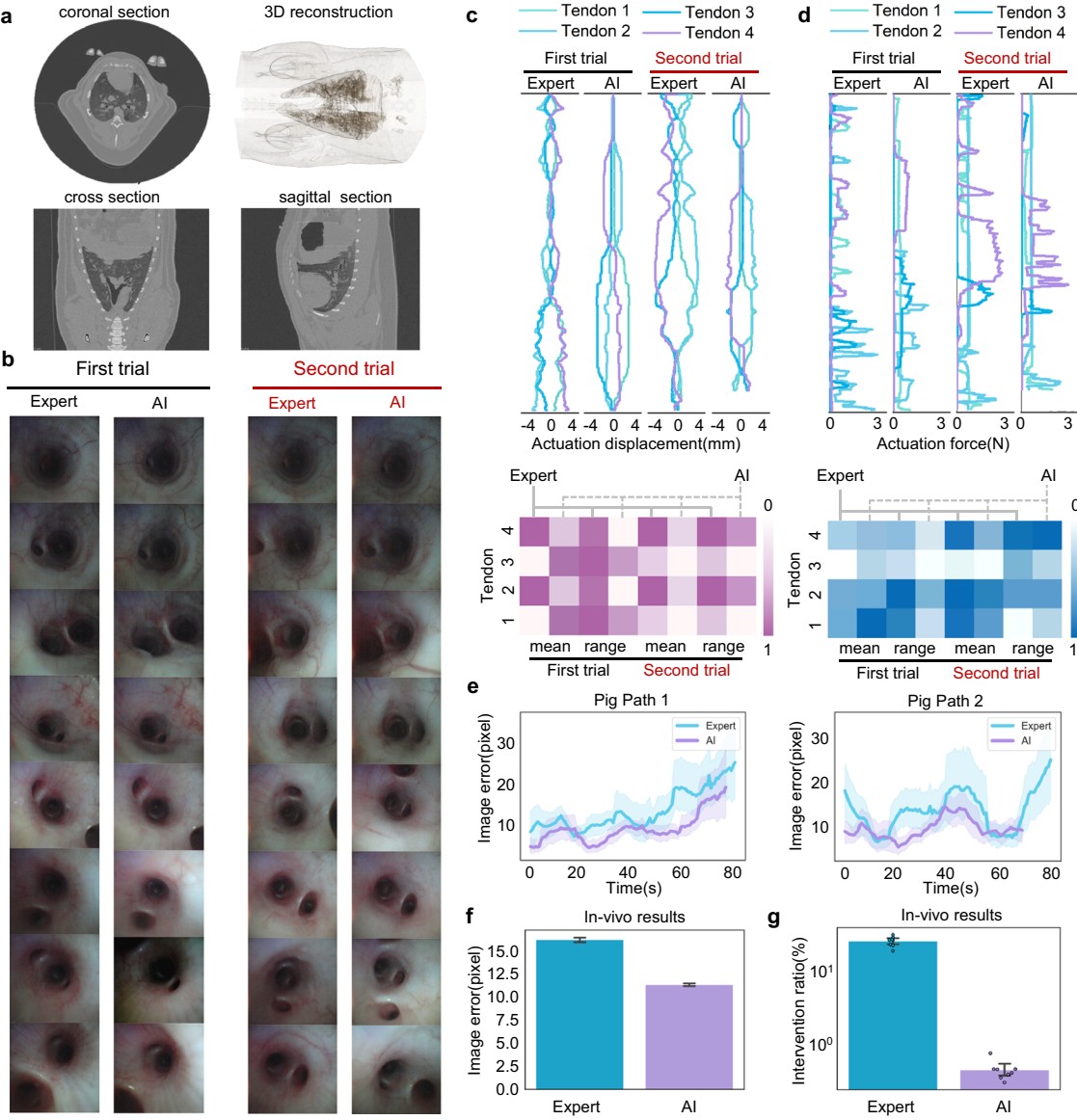

**Fig. 5 | Results of in vivo experiments. a** CT images of the live porcine lung. **b** Bronchoscopic images obtained by the expert without the AI co-pilot and the attending doctor with the AI co-pilot, corresponding to the two paths in Supplementary Fig. 24. **c** Actuation displacement curves and heatmaps of the mean values and fluctuation ranges. **d** Actuation force curves and heatmaps of the mean values and fluctuation ranges. **e** Image error during bronchoscopy. The results show that the attending doctor with the AI co-pilot consistently maintained smaller image errors than the expert without the AI co-pilot. Line plots with error bands show time-varying mean image errors ±95% CI within a time window of 20 frames during each procedure. **f** Statistical results for the image errors. The numbers of recorded frames are $n = 18653$ for Expert and $n = 17354$ for AI. Bar plots depict the mean image error and the 95% confidence interval for all frames during the bronchoscopy procedures in the live porcine lung, i.e., the mean ± 95% CI image error for Path 1 and Path 2. The results show the safer steering achieved in AI-assisted bronchoscopy. **g** Statistical results for the human intervention ratios, determined by the number of time stamps at which the action of the doctor's hand changed from the last time stamp relative to the total number of time stamps recorded throughout the whole bronchoscopy procedure. The results demonstrate that AI-assisted bronchoscopy can effectively minimise the physical strain and cognitive burden on doctors. Human intervention ratios of $n = 8$ independent experiments are presented as mean values ± 95% CI, and specific data points are overlaid on the bar plot.

upon tendon actuation for precise steering control and is fed into the deep lungs by means of "follow-the-tip" motion[41,42]. In alignment with this methodology, the proximal section of the catheter is engineered to exhibit a substantially increased stiffness in comparison to the distal section. However, a large bending angle (approaching 180°) of the distal section presents great challenges in effecting smooth feed movement of the catheter, particularly when negotiating the upper pulmonary regions. A soft untethered magnetic catheter design has the potential to improve the capabilities of bronchoscopy for deep lung examination and is worthy of study. In addition, it is essential to assess the robustness of the proposed method in a broader range of clinical scenarios, including patients with varying bronchial anatomies, pathologies, or respiratory conditions. Extensive testing on a diverse patient population will be necessary to validate the applicability of the intelligent bronchoscope robot in real-world settings. Considering the difference in teleoperators between our AI co-pilot system and existing robotic or hand-held bronchoscopy systems, the relevance between the previous experience of doctors in current teleoperators and the proficiency of operating our system is worth further studying. The integration of additional sensing modalities, such as ultrasound or optical coherence tomography, can also be considered to provide complementary information to guide the bronchoscope robot. Fusing multiple data sources could improve the accuracy and safety

of AI-assisted steering, offering more comprehensive diagnostic and therapeutic support.

In addition, the explainability of our AI co-pilot system was investigated by analysing the reasons for the decision-making of the AI during bronchoscopic procedures. We conducted an experiment on the interpretability of the proposed policy network using three styles of image pairs. We generated gradient-weighted class activation maps (GradCAM) from the last convolutional layer of the policy network to represent the network's attention and visualised the fused images by overlaying the GradCAM results onto the original images. In the resulting images, highlighted regions indicated the key clues that our policy network paid attention to when making decisions. As shown in Supplementary Fig. 25, our network has learned to focus on the bronchial lumens, and as the distance between the robot and the bronchial wall increases, the attention value becomes larger. This indicates that our network concentrates on the structural information of the airway and utilises it to predict safe actions, keeping the centre of the bronchial lumen at the centre of the image. As a result, our AI co-pilot robot is able to remain centred in the airway and stay as far away as possible from the bronchial wall during bronchoscopic procedures.

In conclusion, our AI co-pilot bronchoscope robot offers a promising avenue for enhancing the quality and consistency of bronchoscopic procedures. The system's robust performance in both simulations and in vivo experiments demonstrates its potential to revolutionise bronchoscopy and empower novice doctors to perform these procedures confidently. Looking forward, we anticipate that our approach can be adapted and applied to other medical procedures requiring precise navigation and manipulation, ultimately improving patient outcomes and reducing health care disparities.

## Methods

### Overview of the AI co-pilot system

Our AI co-pilot bronchoscope robot is divided into two main parts: a hardware system and an AI co-pilot algorithm. At the hardware level, the bronchoscope robot employs tendon-driven mechanics, leveraging four linear motors to precisely steer the bronchoscope catheter and an electric slide for feed movement. Additionally, our robotic system boasts an innovative magnetic adsorption method for rapid replacement of the catheter. At the software level, an AI–human shared control algorithm is designed to steer the robot safely. The core of the algorithm is a policy network, which takes both bronchoscopic images and human commands as inputs to predict steering actions that will control the tip of the bronchoscope robot to remain at the centre of the airway, helping prevent injury to the airway mucosa.

To train the policy network, a virtual environment is created to simulate bronchoscopic procedures and collect training data, and then domain adaptation and randomisation techniques are used to enhance the training samples. The training process involves a novel artificial expert agent for automatic data annotation and does not require human intervention. The generator for domain adaptation is pretrained by using virtual bronchoscopic images and unpaired historical bronchoscopy videos, which are easy to access at hospitals, enabling an annotation-free training stage. With the aid of our AI co-pilot bronchoscope robot, the level of human intervention and the cognitive load imposed on doctors can be significantly reduced compared to traditional teleoperated robots.

### AI–human shared control workflow

The working pipeline of our AI–human shared control algorithm in practical use is described as follows. The bronchoscope robot acquires bronchoscopic images during the procedure, and a doctor gives discrete human commands (e.g., left, down, right, up or forward) to determine the high-level direction of the robot. The bronchoscopic images and human commands are input into a trained policy network to predict continuous steering actions (i.e., rotation angle rates $\Delta\theta$ and

$\Delta\varphi$) to control the robot's head such that it remains centred at the bronchial lumen for safety. The predicted steering actions are converted into continuous tendon displacements of the linear motors through inverse kinematics and a low-level controller, forming a closed-loop control system.

### Policy network architecture

The policy network is designed with a multi-task structure, where the main task is steering action prediction and the side task is depth estimation. The learning of the depth estimation task alongside the main task can encourage the network to recognise the 3D bronchial structure and learn a more generalised scene representation for decision-making. The policy network takes a bronchoscopic image (**I**) and a human command ($c$) as inputs, and its outputs include the predicted steering action and estimated depth. Its architecture features an image feature extractor $\Phi_E$, a depth decoder $\Phi_D$ and five branched action heads $\{\Phi_A^i\}_{i=1}^5$, each of which is responsible for predicting steering actions in response to one of five human commands (left, right, up, down and forward). $\Phi_E$ is based on ResNet-34, and $\Phi_D$ is built on a transposed convolutional network, which has skip connections with $\Phi_E$. The action heads are based on multilayer perceptrons (MLPs) and are optionally activated by the human command $c$ through a five-way switch. The depth decoder and action heads share the same representation extracted by the feature extractor. For alignment with the input channels of the MLPs, the features extracted from $\Phi_E$ are flattened to a 512-d vector before being input into the chosen action head. The specific architecture of the policy network is summarised in Supplementary Table 3.

### Training strategy

For training the policy network, a virtual bronchoscopy environment is established based on the segmented airway from preoperative thorax CT scans, as introduced in detail in Supplementary Note 4. In this study, we employ an imitation learning framework to train the policy network. Given an expert policy $\boldsymbol{\pi}^*$, a dataset $D$ of state–command–action triples ($\mathbf{s},c,\mathbf{a}^*$) can be created by executing $\boldsymbol{\pi}^*$ in the virtual bronchoscopy environment. $\mathbf{s}$ represents the state of the environment, which corresponds to the image observed through the bronchoscope robot's camera. $c$ denotes the human command, and $\mathbf{a}^* = \boldsymbol{\pi}^*(\mathbf{s},c)$ represents the expert steering action. The objective of imitation learning is to train a policy network $\boldsymbol{\pi}$, parameterised by $\boldsymbol{\theta}$, that maps any given $\mathbf{s}$ and $c$ to a steering action $\mathbf{a}$ that is similar to the corresponding expert action $\mathbf{a}^*$. By minimising a loss function $\mathcal{L}_a$, the optimal parameters $\boldsymbol{\theta}^*$ can be obtained as follows:

$$\boldsymbol{\theta}^* = \arg\min_{\boldsymbol{\theta}} \sum_{i}^{N} \mathcal{L}_a\left(\boldsymbol{\pi}^*(\mathbf{s_i},c_i),\boldsymbol{\pi}(\mathbf{s_i},c_i;\boldsymbol{\theta})\right) \qquad (1)$$

where $N$ is the size of dataset $D$.

In the conventional imitation learning framework, the expert policy $\boldsymbol{\pi}^*$ is executed by human experts in the environment to collect expert data for training; however, this process is excessively time consuming in practice. In addition, when a behaviour cloning strategy is used to train the policy $\boldsymbol{\pi}$, cascading error and distribution mismatch problems may occur in the inference stage. In our work, an artificial expert agent (AEA) is designed to simulate a human expert and automatically execute the expert policy in the virtual bronchoscopy environment, thereby providing the human command $c$ and annotating the ground-truth expert action $\mathbf{a}^*$ for state $\mathbf{s}$. Thus, the demonstration burden on human experts can be eliminated. We choose the dataset aggregation algorithm DAgger as the imitation learning strategy. The initial dataset is constructed by placing the camera sequentially at waypoints along the centreline and labelling the ground-truth actions and commands obtained from the AEA. A supplementary dataset is then obtained by running the policy network $\boldsymbol{\pi}$ in the virtual

environment and generating frame-by-frame labels with the AEA, namely, the on-policy training process. In the training stage, we choose the L2 loss to implement the action loss, as follows:

$$\mathcal{L}_a(\mathbf{a_i}, \mathbf{a_i^*}) = \frac{1}{N} \sum_{i=1}^{N} ||\mathbf{a_i} - \mathbf{a_i^*}||_2^2 \qquad (2)$$

For depth estimation, a ground-truth depth $\mathbf{d}^*$ can be rendered corresponding to the current observation $\mathbf{s}$, accordingly, the depth loss can be computed as

$$\mathcal{L}_{depth}(\mathbf{d}, \mathbf{d}^*) = \frac{1}{NM} \sum_{i=1}^{N} \sum_{j=1}^{M} ||\mathbf{d_{ij}} - \mathbf{d_{ij}^*}||_2^2 \qquad (3)$$

where $N$ is the size of the whole dataset, $M$ is the number of pixels of each depth, and $\mathbf{d}$ is the estimated depth of the policy network. In the training process, each rollout of a bronchoscopy procedure is terminated by a series of ending conditions, which are described in Supplementary Note 6.

## Artificial expert agent

This section introduces the process of human command generation and ground-truth expert action annotation by the artificial expert agent (AEA). During the training phase, a substantial number of rollouts of virtual bronchoscopy procedures should be performed with human commands, and numerous steering actions must be labelled to ensure adequate samples for training the policy network. This task is labour-intensive and time-consuming for doctors, and the consistency of the resulting human annotations cannot be guaranteed. To address this challenge, we introduce the AEA to automatically provide human commands and annotate ground-truth steering actions based on privileged robot pose information and reference airway centrelines.

As shown in Supplementary Fig. 5b, the ground-truth steering action $[\Delta\theta^*, \Delta\varphi^*]$ is calculated as follows:

$$\Delta\theta^* = \arccos\left(\frac{\overrightarrow{\mathbf{O_cP_a}} \cdot \mathbf{z}}{\overrightarrow{\mathbf{O_cP_a'}}}\right) \qquad (4)$$

$$\Delta\varphi^* = \arcsin\left(\frac{\overrightarrow{\mathbf{P_aP_a'}}}{\overrightarrow{\mathbf{O_cP_a}}}\right) \qquad (5)$$

where $\mathbf{P_a}$ is the target waypoint on the centreline that the robot should be directed towards in the next step, $\mathbf{O_c}$ is the origin of the camera coordinate system, and $\mathbf{P_a'}$ is the projection point of $P_a$ on the $\mathbf{xO_cy}$ plane. $\mathbf{P_a}$ can be determined from the current position of the robot and a fixed distance $d_a$ along the centerline. First, the nearest waypoint $P_n$ on the centreline from the robot's head is selected. Then, $idx_{Pa}$, i.e., the index of $\mathbf{P_a}$ among all waypoints on the centreline, can be calculated as

$$idx_{Pa} = \arg\min_m \left| \sum_{k=n}^{m} \overline{\mathbf{P_kP_{k+1}}} - d_a \right| \qquad (6)$$

where $\mathbf{P_k}$ para_denotes a certain waypoint that lies on the centerline and $\overline{\mathbf{P_kP_{k+1}}}$ is the distance between $\mathbf{P_k}$ and its neighbour $\mathbf{P_{k+1}}$. Thus, the ground-truth steering action $[\Delta\theta^*, \Delta\varphi^*]$ can be annotated for training the policy network.

The principle of human command generation is based on the fact that doctors consistently consider both a far navigation target and a near steering target during bronchoscopy procedures. The far navigation target allows the doctor to assess the risks of the upcoming operation and decide where needs to be examined. The near steering target ensures that the bronchoscope remains at the centre of the airway as much as possible for local safety. The far navigation target

may be approximate yet correct, signifying the desired location the bronchoscope should reach in the near future, similar to the human command in our policy network. For instance, at the junction of the primary and secondary bronchi, the doctor should decide where to examine in the near future. The policy network receives an approximate human command (left or right) as input and generates precise safe steering actions for controlling the robot.

Thus, in the AEA, the human command is determined based on a far target waypoint $\mathbf{P_f}$ and the robot's current position. The index of $\mathbf{P_f}$ can be computed as follows:

$$idx_{Pf} = \arg\min_m \left| \sum_{k=n}^{m} \overline{\mathbf{P_kP_{k+1}}} - d_f \right| \qquad (7)$$

where $d_f$ is the length of the centreline between $\mathbf{P_n}$ and $\mathbf{P_f}$, satisfying $d_f > d_a$. After that, $\mathbf{P_f}$ is projected into the image coordinate system with the known intrinsic parameters of the camera to generate the 2D projected point $\mathbf{P_f}$. The discrete human command $c$ can be computed as

$$c = \begin{cases} forward, 0° \le \angle\mathbf{P_fO_cz} \le \tau \\ up, \angle\mathbf{P_fO_cz} > \tau \cap 45° < \angle\mathbf{p_fOx} \le 135° \\ down, \angle\mathbf{P_fO_cz} > \tau \cap 135° < \angle\mathbf{p_fOx} \le 225° \\ left, \angle\mathbf{P_fO_cz} > \tau \cap 225° < \angle\mathbf{p_fOx} \le 315° \\ right, \angle\mathbf{P_fO_cz} > \tau \cap (0° < \angle\mathbf{p_fOx} \le 45° \cup 315° < \angle\mathbf{p_fOx} \le 360°) \end{cases} \qquad (8)$$

where $\mathbf{O}$ is the origin of the image coordinate system and $\tau$ is the threshold angle of the forward cone for deciding whether to continue forward in the current airway. The five discrete human commands that can be generated by the AEA are encoded as one-hot vectors for input into the policy network.

In practice, the input AEA-annotated human commands are replaced with the doctor's commands in the real bronchoscope robot, driving the policy network to safely and smoothly pass through the airway. The human commands are mapped to five regions of the teleoperator (Supplementary Fig. 3b), reducing the doctor's cognitive load compared to that imposed by conventional teleoperated robots with continuous human intervention.

## Sim2Real adaptation

**Domain adaptation and training strategy.** To improve the performance of the policy network in clinical scenarios, domain adaptation is necessary to reduce the gap between the simulated and real environments. Generative adversarial networks (GANs), which are often used in computer vision for image domain adaptation, can serve our purpose. The generator $\mathbf{G}$ of such a GAN attempt to generate realistic-style images from simulated images, while the discriminator $\mathbf{D}$ attempts to distinguish between generated and real samples. Notably, in clinical scenarios, it is still challenging to pair every bronchoscopic video frame with simulated images rendered from CT airway models due to limited available manpower and the significant visual divergence between body and CT images. When only unpaired data are used for training, existing unpaired image translation methods, such as Cycle-GAN, often misinterpret crucial structural information of the bronchus as part of the style to be translated, leading to inaccurate structures in the generated images.

To address these issues, we propose a structure-preserving unpaired image translation method leveraging a GAN and a depth constraint for domain adaptation. As shown in Supplementary Fig. 9, the network consists of a generator, a discriminator and a depth estimator. Sim-style images rendered from airway models with pink textures are collected to represent the source domain, and their corresponding depths are rendered to provide depth supervision.

Unpaired clinical images from historical bronchoscopic videos, which are easy to access at hospitals, serve to represent the target domain. In the training stage, Sim-style images **x** are fed into the generator to translate them into paired realistic-style images **G(x)**. Then, the discriminator takes both the translated realistic-style images and the unpaired clinical images **y** as input. The adversarial loss is formulated as

$$\mathcal{L}_{GAN}(\mathbf{G},\mathbf{D},\mathbf{x},\mathbf{y}) = \mathbb{E}_{\mathbf{y} \sim p_{data}(\mathbf{y})}[\log \mathbf{D}(\mathbf{y})] + \mathbb{E}_{\mathbf{x} \sim p_{data}(\mathbf{x})}[\log(1 - \mathbf{D}(\mathbf{G}(\mathbf{x})))] \tag{9}$$

Following image translation, the realistic-style images are fed into the depth estimator for the generation of estimated depths. The depth estimation task can be supervised by the rendered depths corresponding to the input rendered images, ensuring that the 3D structure information of each generated image remains consistent with that of the original rendered image. The depth constraint is provided by the depth loss, which is expressed as

$$\mathcal{L}_{\text{depth}}(\mathbf{d},\mathbf{d}^*) = \frac{1}{N} \sum_{i=1}^{N} \left\| \mathbf{d_i} - \mathbf{d_i^*} \right\|_2^2 \tag{10}$$

where $N$ is the number of pixels in the depth image, **d** is the predicted depth and $\mathbf{d}^*$ is the corresponding rendered depth of the input rendered image. As shown in Supplementary Table 4, the backbone of our generator is based on the architecture of AttentionGAN, which explicitly decouples the foreground and background of an image through the introduction of a self-attention mechanism and has shown state-of-the-art performance in recent image translation tasks. The generator **G** is composed of a parameter-sharing feature extractor $\mathbf{G_e}$, an attention mask generator $\mathbf{G_a}$ and a content mask generator $\mathbf{G_c}$. The discriminator is based on the architecture of CycleGAN. The depth estimator comprises a ResNet-34 network for feature extraction and a transposed convolutional network for feature decoding. Specific network architectures of our generator and discriminator can be found in Supplementary Table 4.

### Domain randomisation

To improve the generalisation ability of the policy network, several domain randomisation techniques are designed to randomly alter image appearances or add noise to human commands.

(i) Roll rotation: Since the bronchoscope robot's roll angle is set to 0 in the virtual environment (Supplementary Note 4), we randomly rotate the airway model's roll angle for each rollout. This prevents overfitting of the policy network on limited pulmonary postures and encourages the learning of a more general safe steering policy, regardless of environmental changes. For each rollout, the starting point is randomly placed within the first third of the reference path to collect more challenging data from the deep, thin bronchus, as the trachea and main bronchus account for a large proportion of the entire path.

(ii) Light intensity: To improve the policy network's robustness against variations in light intensity between simulated and clinical scenarios, we randomly adjust the light intensity of the virtual bronchoscopy environment for each observation.

(iii) Command disturbance: Humans may make mistakes when operating robots in practice, such as continuing to control the robot's heading towards the bronchial wall even once a collision has occurred, potentially causing damage or even perforation. To address this, we randomly add disturbances to the human commands before they are input into the policy network when the robot is less than 1 mm away from the bronchial wall during training. This ensures that the policy network will prioritise safe steering over erroneous human control, enhancing safety and reducing the doctor's cognitive load.

(iv) Image attributes: To further improve the generalisation ability of the policy network, we employ four data augmentation methods during the training process to randomly change various attributes of the input images, including brightness, contrast, saturation and hue.

### Implementation details

The AI-human shared control algorithm is implemented using Python (v3.7.11). To establish the virtual bronchoscopy environment, the airway models are segmented from CT scans using 3D Slicer (v4.10.2). The airway centrelines are then extracted using VMTK (v1.4.0). For robot simulation, data acquisition and human command generation, the implementation tools are Pyrender (v0.1.45) and PyBullet (v3.2.2). The policy network and Sim2Real adaptation module are implemented using the PyTorch platform (v1.9.1) and trained on an NVIDIA GTX 2080Ti GPU. The learning rate is set to $10^{-4}$ for training the policy network and $2 \times 10^{-4}$ for training the Sim2Real adaptation module. The batch size for training is set to 64, and both networks are trained for 300 epochs. The acquired images from the simulated and real camera have a size of 400×400, and they are resized to 200×200 before being inputted into the network. The data processing tools include NumPy (v1.19.5), OpenCV (v4.5.5.64) and VTK (v8.2.0). The data analysis and visualisation are implemented using Matplotlib (v3.3.4) and MATLAB R2022a.

### Reporting summary

Further information on research design is available in the Nature Portfolio Reporting Summary linked to this article.

## Data availability

The data used for establishing virtual bronchoscopy environment and training networks are available at https://zenodo.org/records/10077275, whereas the trained network model data used in simulated, in-vitro and in-vivo experiments is available at https://zenodo.org/records/10077290. Other data needed to evaluate the conclusions are provided in the main text, and Supplementary files. Source data are provided with this paper.

## Code availability

The code is available on GitHub (https://github.com/LiuLiluZJU/AI-Co-Pilot-Bronchoscope-Robot)[43]. The DOI for the code is https://doi.org/10.5281/zenodo.10077315. The repository includes virtual environment establishment, data acquisition, image processing, visualisation, network training and testing code.

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

## Acknowledgements

This work was supported by the National Key R&D Program of China under Grant no. 2021ZD0114500 (R.X.); the National Natural Science Foundation of China under Grants no. 62373322 (Y.W.), no. T2293724 (H.L.) and no. 62303407 (H.L.); the Key R&D Program of Zhejiang under Grants no. 2023C01176 (Y.W.) and no. 2022C01022 (H.L.); and the Zhejiang Provincial Natural Science Foundation of China under Grant no. LD22E050007 (H.L.).

## Author contributions

J.Z. designed and fabricated the whole bronchoscope robot. L.L. designed the AI–human shared control algorithm. J.Z., L.L. and P.X. developed the software and carried out all the experiments. Q.F. and X.N. validated the work. H.M. and J.H. provided technical support for the clinical experiments. R.X., Y.W. and H.L. provided technical support for the algorithm design and funding support. J.Z. and L.L. wrote the paper.

## Competing interests

The authors declare no competing interests.
