## [Peer Review File · Nature Communications]

AI Co-pilot Bronchoscope RobotREVIEWER COMMENTS

Reviewer #1 (Remarks to the Author):

Thank you for the opportunity to review this submission from Zhang et al. The authors describe a novel robotic bronchoscopy platform with AI co-pilot mean to augment control and stabilize visualization of the bronchial lumen during procedures performed by novice users. The manuscript contains a significant amount of data showing how the system can improve performance. I have several comments:

General comments:

The manuscript is highly technical and contains a wealth of formulaic information. However, for this reason it is also difficult to comprehend by those without extensive experience in this area. Also, while it contains a number of “supplementary notes”, it has no Methods section that I can identify. This results in a lack of critical definitions. For instance, who is considered an “experienced user”? Is this a thoracic surgeon? Pulmonologist? How many bronchoscopy procedures have they performed? Who are the novice users? Do they have any bronchoscopy experience? In practical terms, what does a 50 pixel image error mean? What is actuation displacement? What are the primary outcomes as measured in the study?

The introduction suggests that this technology might be useful in “under-privileged” or resource-limited areas where qualified providers are not available. However, the system and technology seem like they would be fairly costly. Is this practical for such resource-limited institutions to acquire? If not, this technology could still have value for providing technical improvement even for experienced users.

Line 253 suggests that the AI integration “greatly reduces the doctor’s physical exertion and cognitive load during bronchoscopy operation.” This statement does not appear to be well supported by the data. Primarily this data is from Figure 5g which demonstrates a reduced “intervention ratio”. Is there evidence that this is directly related to physical fatigue or cognitive strain? Usually these can be measured with other tools such as the NASA-TLX survey.

Comments about the bronchoscopy system:

It is unclear from the manuscript if this is a guided bronchoscopy platform or if it strictly a mechanized bronchoscopy device with enhanced stabilization. Does it use electromagnetic navigation, CT planning or

other features to provide a pathway to the lesion? If so, how does the hardware integrate pre-procedural planning information?

A 1.1 working channel seems quite small. Most tools (needles, forceps, etc) are at least 1.5mm in diameter. What applications do the authors envision the platform being useful for? Strictly lavage?

What is the degree of flexion and bend radius of the catheter? From the video it looks like almost 180 degrees but I don't think this is stated in the manuscript.

Also, it is unclear how the system is controlled. Perhaps some photos or a video of the controller during the procedure would be helpful.

Comments regarding Figures:

Figure 2 is very dense and hard to follow.

Figure 3 – Figure labels could be improved. For instance, how does figure 3c differ from 3d? In the text it sounds as though 3d is related to path length whereas 3C is related to simply correct vs. incorrect path. This should be clear from the figure.

Figure 4 – for d and e, what is depicted in the bar plots? Is it average image error over the duration of the procedure?

Figure 5 – in c and d, which actuation displacement curve corresponds to which condition? Is it the same order as 5b?

Supplemental Figure 11 – for a and b, the figure legend mentions 5 groups but to my eye there are only 4 sets of curves. This should be clarified.

Supplemental Figure 14 – a through d. These conditions need to be better labelled. 14a for instance talks about data from experiments in 2 phantoms from 1 user but there are 4 sets of 4 curves.

Movies – The movies are interesting however several of the videos are very blurry and the zip files takes over an hour to download.

Reviewer #2 (Remarks to the Author):

Please see the attached file.

This work proposes an AI policy network for guiding a robot during bronchoscopy. The AI shares control with a surgeon so provide safer motions when navigating through the lungs. The authors' tested their robot in both in-vitro and in-vivo settings with different levels of experienced surgeons. This paper would be of interest in the field of surgical robotics and machine learning. The focus is mainly on the implementation of an AI software as a method for sharing control of the robot with a human. The methods used by the authors result in a novel contribution to the field. A few questions and considerations remain when reading this work that would help strengthen and clarify the conclusions.

In the introduction, the authors make valid claims about the potential issues that may arise in traditional bronchoscopy. Adding some references, if available, to quantify the frequency or severity of these issues would help better frame the proposed robot with respect how the procedure is currently performed.

While few state-of-the-art robot bronchoscopes are mentioned, there should be more detail to explain their relevance with respect to the proposed work. Also, there are more state-of-the-art bronchoscopy robots being created in research labs that have shown new tools for bronchoscopy, implementations of control, and safe navigation. Adding greater detail and more references to the state-of-the-art would help better frame the authors' work. The authors can see this list of references as a starting point for bronchoscopy robots that are currently being researched:

1. Bao, Y., Li, X., Wei, W., Liu, H., Qu, S. (2022). Study on the interventional path planning method of bronchoscope catheter. *Journal of Mechanical Science and Technology* 2022 36:5, 36(5), 2643–2652.
2. Van Lewen, D., Janke, T., Lee, H., Austin, R., Billatos, E., Russo, S. (2023). A Millimeter-Scale Soft Robot for Tissue Biopsy Procedures. *Advanced Intelligent Systems*, 2200326.
3. Pittiglio, G., Lloyd, P., da Veiga, T., Onaizah, O., Pompili, C., Chandler, J.H. and Valdastrri, P., 2022. Patient-specific magnetic catheters for atraumatic autonomous endoscopy. *Soft Robotics*, 9(6), pp.1120-1133.
4. McCandless, M., Perry, A., DiFilippo, N., Carroll, A., Billatos, E., Russo, S. (2022). A Soft Robot for Peripheral Lung Cancer Diagnosis and Therapy. *Soft Robotics*, 9(4), 754–766.

The idea of centering the bronchoscope within the lung branch during navigation will benefit the safety of the patient. In general, navigating a bronchoscope requires some interaction with the lung walls when making turns. The robot may not be able to stay centered along the full length it has traveled. Will the task of staying centered at the distal tip concentrate extra stress on the lung walls elsewhere along the robot's body?

It is unclear if the AI co-pilot is only predicting the steering actions and displaying visual feedback to the surgeon or if it is also applying some intervention and modifying the steering actions of the surgeons. If the latter is the case, would the surgeon be able to take control of the robot from the AI co-pilot if they wanted?

In the simulation experiments, the success rate is defined as ratio of successful paths to all paths. The criteria for a successful path was not explained. Does this refer to mapping reference paths to the simulated bronchoscope images? If this is the case, how is the simulated path derived from the simulated bronchoscope images? The same issue is found for the successful path ratio. What is the criteria for a path to be completed? Further, explanation of the simulation setup and success criteria would help in understanding.

The in-vitro evaluation procedure with the inclusion of breathing motion is well described and is setup well with both novice and experienced surgeons. The results of both in-vitro and in-vivo experiments are reported in the form of image errors (pixels). It is unclear what the image errors translate to a clinical scenario. Is this a distance, in pixels, between the centers of each image? If this is the case, is image error a relevant benchmark when the difference in error between the AI and expert are on the order of 10 pixels? At such a small scale in bronchoscopy, this difference in image error likely translates to less than a millimeter of error depending on the camera specifications. Further details on this would be helpful in understanding these results. Also, the paths chosen for these evaluations seem to be straight-forward with less steering required. Is the robot able to make tighter bends into the upper portions of the lung?

Familiarity with teleoperation can make a difference in surgeon skills. Have the surgeons used a similar teleoperation platform before these trials were performed? If not, would allowing the surgeons time to train on

the teleoperation platform change the results? Any details on this would strengthen the comparison to an expert surgeon since an expert surgeon with no experience in teleoperation may lead to results showing greater error than if the surgeon had teleoperation experience.

Figure 1 shows the bronchoscope robot; however, the robot components are not well explained in the main text. What kind of microcamera and materials are used to make the robot? This would help with reproducibility of the proposed work. Further, the authors claim that this robot will address disparities in healthcare; however, the proposed robot does not seem to make progress toward accessibility of bronchoscopy procedures in underdeveloped regions based on the types of materials.

How do the dynamics of the bronchoscope robot play a role in controlling and keeping the bronchoscope centered? In some of the movies, there seems to be overshooting in the AI system, will this be safe? Furthermore, does the use of only five discrete commands limit the motion of the robot to two planes or can the robot be bent with multiple tendons being pulled at a time? An evaluation of the overshooting effect and clarification of the control strategy would be helpful in understanding this.

Is the robot arm that holds the bronchoscope robot also controlled with the AI co-pilot and human teleoperator or is the robot arm set in a fixed position throughout the whole procedure? Could a different robot arm position result in a larger image error in the experiments? How does the AI co-pilot affect the overall speed of the procedure?

In the in-vivo experiments, the number of interventions is reported as a result which characterizes autonomy. However, it is stated that the number of interventions is lower than those interventions of the expert. This statement seems counterintuitive since there is constant human intervention in the case of the expert surgeon. A definition of human intervention by the authors would be helpful here.

Reviewer #3 (Remarks to the Author):

Thank you for giving me the opportunity to review this very interesting manuscript.

I have some comments.

In Introduction: please, improve the state of the art by citing the most relevant articles on Monarch Platform and Ion Endoluminal System.

On line 125: please add diameter of Monarch Platform and Ion Endoluminal System.

How many novice doctors tested your AI robot?

The Discussion is too short. It should be improved.

Please, add some context from the evidence of published literature on Monarch Platform and Ion Endoluminal System.

I suggest adding some paragraphs to frame your work in terms of explainable AI.

What about the limitations of your work?

In Supplementary Note 4.

I suggest adding details on how the segmentation was performed.

Response to Reviewers' Comments on "NCOMMS-23-26020-T"

We would like to thank the reviewers for their constructive remarks and input that help us further improve the manuscript. Each comment from the editor and reviewers has been carefully considered and addressed, with additional experiments. Point-to-point responses to each comment are listed in the following, with corresponding changes highlighted in the revised manuscript in blue for easy tracking.

Reviewer #1

Thank you for the opportunity to review this submission from Zhang et al. The authors describe a novel robotic bronchoscopy platform with AI co-pilot mean to augment control and stabilize visualization of the bronchial lumen during procedures performed by novice users. The manuscript contains a significant amount of data showing how the system can improve performance.

Response to Comment:

We thank the referee very much for careful reading of our manuscript and valuable comments, which helps to improve the quality of our manuscript. We are pleased that the referee finds our work is novel and the data is sufficient. Following the referee's comments and suggestions, we have revised our manuscript with the most seriousness and please find our detailed responses below.

General comments:

Comment 1:

The manuscript is highly technical and contains a wealth of formulaic information. However, for this reason it is also difficult to comprehend by those without extensive experience in this area. Also, while it contains a number of "supplementary notes", it has no Methods section that I can identify.

Response to Comment 1:

We appreciate the reviewer for pointing out the readability issue of our paper. We apologize for not providing an obvious Method section in our previous version of manuscript. For a more comprehensive understanding of the technical intricacies, we have added a Method section in our main text to provide detailed technical aspects.

Revised content, on page 10-16 of the main text:

Method

AI co-pilot system overview

AI co-pilot bronchoscope robot is divided in two main parts: hardware system and AI co-pilot algorithm. In hardware level, the bronchoscope robot employs tendon-driven mechanics, leveraging four linear motors to precisely steer the bronchoscope catheter, and an electric slide for feed movement. Additionally, our robot system boasts an innovative magnetic adsorption method for rapidly replacement of the catheter.

Detailed characterization analysis is depicted in Supplementary Note 1-3. In software level, an AI-human shared control algorithm is designed to steer the robot safely. The core of the algorithm is a policy network, which takes both bronchoscopic image and human command as input to predict steering action, which controls the tip of bronchoscope robot staying at the center of airway and helps prevent injury to airway mucosa.

For training the policy network, a virtual environment is created to simulate bronchoscopy procedure and collect training data, then domain adaptation and randomization techniques are used to enhance training samples. The training process involves a novel artificial expert agent for automatic data annotation and does not require human intervention. The generator of domain adaptation is pre-trained by using virtual bronchoscopic images and unpaired historical bronchoscopy videos, which are easy to access in hospital and annotation-free in training stage. With the aid of our AI co-pilot bronchoscope robot, human intervention and cognitive load of doctors can be significantly reduced compared to traditional teleoperated robots.

AI-human shared control workflow

The working pipeline of our AI-human shared control algorithm in practical use is described as follows. The bronchoscope robot takes bronchoscopic image during procedure, and a doctor gives discrete human command (e.g., left, down, right, up or forward) to decide the high-level direction of robot. Bronchoscopic image and human command are inputted in to a trained policy network to predict continuous steering actions (i.e. rotation angle rates $\Delta\theta$ and $\Delta\varphi$) to control the robot's head centered at the bronchial lumen for safety. Predicted steering actions are converted to continuous tendon displacements of linear motors by inverse kinetics and low-level controller, forming a closed loop control system.

Policy network architecture

The policy network is designed as a multi-task structure, where the main task is steering action prediction and the side task is depth estimation. The learning of depth estimation task alongside can encourage the network to recognize the bronchial 3D structure and learn a more generalized scene representation for decision-making. The policy network takes as inputs a bronchoscopic image (I) and a human command (c), with its outputs including the predicted steering action and estimated depth. Its architecture features an image feature extractor Φ_E , a depth decoder Φ_D and five branched action heads $\{\Phi_A^i\}_{i=1}^5$ responsible for predicting steering actions in response to five human commands (left, right, up, down and forward) respectively. Φ_E is based on the ResNet-34 and Φ_D is built on the transposed convolutional network, which has skip connections with Φ_E . Action heads are based on multilayer perceptron (MLP) and can be optionally activated by the human command c through a five-way switch. The depth decoder and action heads share the same representation extracted by feature extractor. For alignment with the input channel of MLP, the feature extracted from Φ_E is flattened to a 512-d vector before inputting into the chosen action head. The specific architecture of the policy network is shown in Supplementary Table 2.

Training strategy

For training the policy network, a virtual bronchoscopy environment is established by the segmented airway from pre-operative thorax CT scans, which is detailedly introduced in Supplementary Note 4. In this study, we employ an imitation learning framework to train the policy network. Given an expert policy π^* , the dataset D of state-command-action pairs (s, c, a^*) can be created by executing π^* in virtual bronchoscopy environment. The s represents the state of the environment, which is the observed image through the bronchoscope robot’s camera. The c denotes the human command, and $a^* = \pi^*(s, c)$ represents the expert’s steering action. The objective of imitation learning is to learn a policy network π , parameterized by θ , that maps any given s and c to a similar steering action a to expert’s action a^* . By minimizing a loss function \mathcal{L}_a , the optimal parameters θ^* can be obtained as

$$\theta^* = \arg \min_{\theta} \sum_i^N \mathcal{L}_a(\pi^*(s_i, c_i), \pi(s_i, c_i; \theta)) \quad (1)$$

where N is the size of dataset D .

In conventional imitation learning framework, the expert policy π^* is executed by human experts in the environment to collect expert data for training, which is time-consuming in practice. Besides, when using behavior cloning strategy to train the policy π , the cascading error and distribution mismatch problems may occur in inference stage. In our work, the artificial expert agent (AEA) is designed to simulate human expert and automatically execute the expert policy in the virtual bronchoscopy environment, which can provide human command c and annotate the ground truth expert’s action a^* for state s . Thus, the demonstration burden of human experts can be eliminated. We choose Dataset Aggregation (DAgger) algorithm as the imitation learning strategy. The initial dataset is composed by placing the cameras sequentially on waypoints of the centerline and labeling the ground truth actions and commands by AEA. The supplementary dataset is obtained by running policy network π in the virtual environment and labeled frame by frame by the AEA, namely on-policy training process. In training stage, we choose L2 loss to implement the action loss as

$$\mathcal{L}_a(a_i, a_i^*) = \frac{1}{N} \sum_{i=1}^N \|a_i - a_i^*\|_2^2 \quad (2)$$

For depth estimation, a ground truth depth d^* can be rendered corresponding to current observation s , thus the depth loss can be computed as

$$\mathcal{L}_{depth}(d, d^*) = \frac{1}{NM} \sum_{i=1}^N \sum_{j=1}^M \|d_{ij} - d_{ij}^*\|_2^2 \quad (3)$$

where N is the size of the whole dataset, M is the number of pixels of each depth, d is the estimated depth of the policy network. In training process, each rollout of bronchoscopy is terminated by a series of ending conditions, which are described in Supplementary Note 6.

Artificial expert agent

This section introduces the process of human command generation and ground truth expert’s action annotation by the artificial expert agent (AEA). During the training phase, a substantial number of rollouts of virtual bronchoscopy should be performed with human commands and numerous steering actions must be labeled to ensure adequate samples for training policy network. This task is labor-intensive and time-

consuming for doctors, and the consistency in human annotations cannot be guaranteed. To tackle this challenge, we introduce AEA to automatically provide human commands and annotate ground truth steering actions, with the privileged robot pose and reference airway centerlines.

As shown in Supplementary Fig. 5b, the ground truth steering action $[\Delta\theta^*, \Delta\varphi^*]$ is calculated as follows:

$$\Delta\theta^* = \arccos\left(\frac{\overline{O_c P_a' z}}{O_c P_a'}\right) \quad (4)$$

$$\Delta\varphi^* = \arcsin\left(\frac{\overline{P_a P_a'}}{O_c P_a'}\right) \quad (5)$$

where P_a is the target waypoint on centerline that robot should direct to in next step, O_c is the origin of the camera coordinate system, P_a' is the projection point of P_a on $xO_c y$ plane. P_a can be determined by the current robot position and a fixed distance d_a along the centerline. Firstly, the nearest waypoint P_n on centerline from robot's head is selected. Then, idx_{P_a} , i.e. the index of P_a among all waypoints on centerline, can be calculated as

$$idx_{P_a} = \operatorname{argmin}_m |\sum_{k=n}^m \overline{P_k P_{k+1}} - d_a| \quad (6)$$

where P_k denotes a certain waypoint that lies on the centerline and $\overline{P_k P_{k+1}}$ is the distance between P_k and its neighbor P_{k+1} . Thus, the ground truth steering action $[\Delta\theta^*, \Delta\varphi^*]$ can be annotated for training the policy network.

The principle of human command generation is based on the fact that doctors consistently possess a far navigation target and a near steering target in mind during bronchoscopy procedures. Far navigation target allows doctors to assess the risks of the upcoming operation and decide where needs to be examined. Near steering target ensures the bronchoscope staying at the center of airway as much as possible for local safety. This far navigation target may be approximate yet correct, signifying the desired location the bronchoscope should reach in the near future, similar to the human command of our policy network. For instance, at the junction of primary and secondary bronchi, the doctor should decide where to exam in near future. The policy network receives an approximate human command (left or right) as input and generates precise safe steering actions for controlling the robot.

Thus, in AEA, the human command is determined by a far target waypoint P_f and current robot's position. The index of P_f can be computed as:

$$idx_{P_f} = \operatorname{argmin}_m |\sum_{k=n}^m \overline{P_k P_{k+1}} - d_f| \quad (7)$$

where d_f is the length of centerline between P_n and P_f , satisfying $d_f > d_a$. After that, P_f is projected to the image coordinate system with known camera intrinsic parameters to generate the 2D projected point p_f . The discrete human command c can be computed as

$$c = \begin{cases} \text{forward, } 0^\circ \leq \angle P_f O_c \mathbf{z} \leq \tau \\ \text{up, } \angle P_f O_c \mathbf{z} > \tau \cap 45^\circ < \angle p_f O \mathbf{x} \leq 135^\circ \\ \text{down, } \angle P_f O_c \mathbf{z} > \tau \cap 135^\circ < \angle p_f O \mathbf{x} \leq 225^\circ \\ \text{left, } \angle P_f O_c \mathbf{z} > \tau \cap 225^\circ < \angle p_f O \mathbf{x} \leq 315^\circ \\ \text{right, } \angle P_f O_c \mathbf{z} > \tau \cap (0^\circ < \angle p_f O \mathbf{x} \leq 45^\circ \cup 315^\circ < \angle p_f O \mathbf{x} \leq 360^\circ) \end{cases} \quad (8)$$

where O is the origin of image coordinate system, and τ is the threshold angle of the forward cone for deciding whether keep forward in current airway or not. Five discrete human commands generated by AEA are encoded as one-hot vectors for inputting into the policy network.

In practice, the input AEA annotated human commands of the real bronchoscope robot are replaced with doctor's commands, driving the policy network to safely and smoothly pass through the airway. The human commands are mapped to five regions of the teleoperator (Supplementary Fig. 3b), reducing doctor's cognitive load compared to conventional teleoperated robots with continuous human intervention.

Sim2Real adaptation

Domain adaptation and training strategy: To improve the performance of the policy network in clinical scenarios, domain adaptation is necessary to reduce the gap between simulated and real environments. Generative Adversarial Networks (GANs), often used in computer vision for image domain adaptation, serve our purpose. The generator G attempts to generate realistic-style images from simulated images, while the discriminator D tries to distinguish between generated and real samples. In clinical scenarios, it's still a challenge to pair every bronchoscopic video frames to simulated images rendered from CT airway models due to the limited manpower and significant visual divergence between body and CT images. When using only unpaired data for training, existing unpaired image translation methods, such as CycleGAN, often misinterpret the crucial structural information of bronchus as a part of style to be translated, leading to inaccurate structures in the generated images.

To address these issues, we propose a structure-preserving unpaired image translation method leveraging GAN and depth constraint for domain adaptation. As shown in Supplementary Fig. 9, the network consists of a generator, discriminator and a depth estimator. The Sim-style images rendered from airway models with pink texture are collected as the source domain, and their corresponding depths are rendered to provide depth supervision. Unpaired clinical images from historical bronchoscopy videos serve as the target domain, which are easy to access in hospital. In the training stage, Sim-style images x are fed into the generator to translate them into paired realistic-style images $G(x)$. Then the discriminator takes both translated realistic-style images and unpaired clinical images y as input. The adversarial loss is formulated as

$$\mathcal{L}_{GAN}(G, D, x, y) = \mathbb{E}_{y \sim p_{data}(y)}[\log D(y)] + \mathbb{E}_{x \sim p_{data}(x)}[\log(1 - D(G(x)))] \quad (9)$$

Following image translation, the realistic-style images are fed into the depth estimator for generating estimated depths. The depth estimation task can be supervised by the rendered depths corresponding to the input rendered image, ensuring that the 3D

structure information of the generated image remains consistent with the original rendered image. The depth constraint is provided by the depth loss as

$$\mathcal{L}_{\text{depth}}(d, d^*) = \frac{1}{N} \sum_{i=1}^N \|d_i - d_i^*\|_2^2 \quad (10)$$

where N is the number of pixels in depth image, d is the predicted depth and d^* is the corresponding rendered depth of input rendered image. As shown in Supplementary Table 1, the backbone of our generator is based on the architecture of AttentionGAN, which explicitly decouples the fore- and background of image by introducing the self-attention mechanism and has shown the state-of-the-art performance in recent image translation tasks. The generator G is composed of a parameter-sharing feature extractor G_e , an attention mask generator G_a and a content mask generator G_c . The discriminator is based on the architecture of CycleGAN. The discriminator is based on the CycleGAN architecture, and the depth estimator comprises a ResNet34 for feature extraction and a transposed convolutional network for feature decoding.

Domain randomization: To improve the generalization capability of policy network, some domain randomization techniques are designed to randomly alters image appearances or adds noise to human commands.

(i) Rotation roll: Since the bronchoscope robot's *roll* angle is set to 0 in the virtual environment (Supplementary Note 4), we randomly rotate the airway model's *roll* angle for each rollout. This prevents policy network overfitting on limited pulmonary postures and encourages learning a more generalized safe steering policy, regardless of environmental changes. For each rollout, the starting point is randomly placed within the first third of the reference path to collect more challenging data in deep, thin bronchus, as the trachea and main bronchus account for a large proportion of the entire path.

(ii) Light intensity: To improve the policy network's robustness against variations in light intensity between simulated and clinical scenarios, we randomly adjust the light intensity of virtual bronchoscopy environment for each observation.

(iii) Command disturbance: Human may mistakenly operate robot in practice, such as constantly controlling robot heading to bronchial wall even if collision has occurred, potentially causing damage or even perforation. To address this, we randomly add disturbances to the human command before it is inputted into policy network, when the robot is less than 1mm away from the bronchial wall during training. This ensures the policy network prioritizes safe steering over erroneous human control, enhancing safety and reducing the doctor's cognitive load.

(iv) Image attributes: To further improve the generalization ability of policy network, we employ four data augmentation methods during the training process by randomly changing input image attributes, including brightness, contrast, saturation, and hue.

Comment 2:

This results in a lack of critical definitions. For instance, who is considered an “experienced user”? Is this a thoracic surgeon? Pulmonologist? How many

bronchoscopy procedures have they performed? Who are the novice users? Do they have any bronchoscopy experience?

Response to Comment 2:

We appreciate the reviewer's thoughtful inquiries. The "experienced user" in this paper is defined as a thoracic surgeon who possess extensive practical experience on performing hand-held bronchoscopy operations. These professionals have undergone training to proficiently operate the bronchoscope robot using teleoperation and AI-human shared control methods, having completed this bronchoscopy procedures over hundreds of iterations. In the submitted manuscript, we utilized the terms "expert" and "experienced doctor" to refer to one kind of medical doctors. Therefore, we have uniformed this nomenclature in our revised manuscript.

The novice user is defined as one kind of doctors, i.e., attending physician and medical intern, who lack significant experience in performing robotic bronchoscopy operations, and have watched the operational videos of the bronchoscope robot.

To underscore the practical application of our research, we have collaborated with an expert, Chief Physician, from the School of Medicine at Zhejiang University in China. The experimental results conducted by the expert are illustrated in the **Fig. 4c, d, e**, and **Fig. 5** in the revised manuscript. In addition, we also engaged two novice doctors including a medical intern and a attending doctor from the School of Medicine at Zhejiang University, Hangzhou, China. The results from the intern are depicted in **Fig. 4c**, while the results from the attending doctor are showcased in **Fig. 4c, d, e**, as well as **Fig. 5**. To demonstrate the operation experience of these participants more intuitively, we added **Supplementary Table 3** in the revised manuscript.

revised content:

Supplementary Table 3. The operation experience of participants in the experiments. The medical intern (novice 1), the attending doctor (novice 2) and the expert doctor (expert) both come from the School of Medicine at Zhejiang University, Hangzhou, China.

Participant	Professional title	Entire period of operation	Manual operations	Robot teleoperations
novice 1	medical intern	no	no	2 demonstrations
novice 2	attending doctor	<5 years	<100 cases per year	2 demonstrations
expert	chief doctor	>20 years	>200 cases per year	>100 trials

Comment 3:

In practical terms, what does a 50-pixel image error mean?

Response to Comment 3:

We apologize for not providing detailed explanation about the definition of image error and the calibration between image pixel and 3D distance. In practical terms, 50-pixel image error means the distance is 50 pixels from image center to bronchial lumen center in the image coordinate system, which reflects approximate 3.20mm 3D position error

between the robot's head and the airway centerline in 3D space. Besides, in our in-vivo animal experiment, AI co-pilot group has a mean image error of 11.38 pixels during all bronchoscopy procedures, which reflects a mean 3D position error of 0.73mm. In this study, the pixel-to-millimeter conversion ratio is about 0.064mm/pixel. Commonly, the accuracy of bronchoscopy procedure is directly measured by the 3D position error, which is the Euclidean distance from the robot's head to the nearest waypoint on the reference path (i.e. airway centerline) in 3D space. However, in practical dynamic scenarios, due to respiratory movement and deformation of bronchus, 3D position error is hard to measure using existing electromagnetic and visual tracking techniques, which prone to introduce a large system error in ground truth reference path localization. Thus, we employ the image error as the metric to measure the accuracy of bronchoscopy procedure, which represents the distance in pixel from image center to bronchial lumen center in the image coordinate system and can also support our conclusions. Specific definition of image error and experiments on pixel-to-millimeter calibration are introduced as follows.

Fig. R1 (Supplementary Fig. 19). Image error to 3D position error mapping. **a**, Definitions of image error and 3D position error. The image error e_{img} is projected by a predicted direction vector $\hat{\mathbf{v}}$ by the policy network, which directs to a point lying on the centerline in 3D space. And the 3D position error e_{pos} is defined as the nearest distance between robot’s head and the reference path. **b**, Statistical results of the pixel-to-millimeter conversion ratio in different bronchial generations. **c**, Virtual environment established on the airway model of Phantom 1 with its two reference paths Path 19 and 55. **d**, Virtual environment based on Phantom 2 airway model with Path 25 and 66. **e**, Segments of bronchial generations along Path 19 and 55 in Phantom 1. **f**, Segments of bronchial generations along Path 25 and 66 in Phantom 2. **g**, Distribution of pixel-to-millimeter conversion ratio along reference paths in Phantom 1. **h**, Distribution of pixel-to-millimeter conversion ratio along reference paths in Phantom 2. It’s obvious that with the bronchoscope robot reaching deep bronchus, the pixel-to-millimeter conversion ratio becomes smaller, because the diameter of airway tree becomes thinner.

Definition of image error: The accuracy of bronchoscopy procedures is commonly measured by the 3D position error, which is the Euclidean distance between the robot’s head and the reference path (i.e. centerline) in 3D space, shown as e_{pos} in **Fig. R1a**. Some existing works have reported position error results (also called absolute tracking error, ATE) of bronchoscopy in dead porcine lung or static phantom by aligning pre-operative reference path with realistic bronchial tree as the ground truth path. The key hypothesis of these works is that the lung is static and rigid when performing bronchoscopy. However, in clinical scenarios, due to the respiratory motion of live lung and the deformation of bronchus, all existing methods fails to accurately align reference paths with realistic bronchial tree, resulting in inaccurate 3D position error measurement, which is especially severe in deep bronchi. To address this issue, in this study, we use image error to measure the accuracy of bronchoscopy procedure, shown as e_{img} in **Fig. R1a**. The image error is a widely used metric in the field of robot visual serving, which measures the distance from image center to the target position in image coordinate system without the need of 3D position. In practical terms, a mapping from e_{img} to e_{pos} can be formed in a statistical way, which we have performed new experiments for discussion in the following part.

The image error e_{img} is calculated by projecting the next direction vector $\hat{\mathbf{v}} \in \mathbb{R}^3$, predicted by the policy network, into the current image coordinate system of robot. In this study, we assume the policy network is well trained and $\hat{\mathbf{v}}$ should point to a waypoint lying on the centerline, satisfying

$$e_{img} = \|\overline{Op}\| = \left\| \frac{1}{\hat{\mathbf{v}}_z} \mathbf{K} \hat{\mathbf{v}}^T \right\| \quad (\text{R1})$$

where \overline{Op} is projected from \hat{v} in image, $\hat{v}|_z$ is the z-coordinate value of \hat{v} , $\mathbf{K} = \begin{bmatrix} f_x & 0 & 0 \\ 0 & f_y & 0 \end{bmatrix}$ is the known intrinsic matrix of camera, f_x and f_y are intrinsic parameters and satisfy $f_x = f_y$. Actually, \hat{v} is predicted by the policy network π and derived from the steering action $a = \pi(x, c) = [\Delta\theta, \Delta\varphi]$. According to the formulation of ground truth steering action in **Fig. 2c**, if the policy network is well trained, \hat{v} will direct to a point P_a on the centerline, and the projected \overline{Op} will direct to the bronchial lumen in image coordinate system. By using steering action, \hat{v} can be rotated from the current direction vector $v = [0, 0, d]$ of robot with a fixed depth d as

$$\hat{v}^T = \mathbf{R}(\Delta\theta, \Delta\varphi)v^T = \mathbf{R}(\Delta\theta, \Delta\varphi) \begin{bmatrix} 0 \\ 0 \\ d \end{bmatrix} \quad (\text{R2})$$

where $\mathbf{R}(\Delta\theta, \Delta\varphi) \in \mathbb{R}^{3 \times 3}$ is the rotation matrix from v to \hat{v} , parameterized by the steering action $[\Delta\theta, \Delta\varphi]$ as

$$\mathbf{R}(\Delta\theta, \Delta\varphi) = \begin{bmatrix} \cos(\Delta\theta) & 0 & \sin(\Delta\theta) \\ \sin(\Delta\varphi)\sin(\Delta\theta) & \cos(\Delta\varphi) & -\cos(\Delta\theta)\sin(\Delta\varphi) \\ -\cos(\Delta\varphi)\sin(\Delta\theta) & \sin(\Delta\varphi) & \cos(\Delta\varphi)\cos(\Delta\theta) \end{bmatrix} \quad (\text{R3})$$

For simplicity, the rotation can also be parametrized by an angle $\psi = f(\Delta\theta, \Delta\varphi)$ about an axis of rotation, as shown in **Fig. R1a**. Thus, we can rewrite Eq. R1 as following

$$e_{img} = \left\| \frac{1}{\hat{v}|_z} \mathbf{K} \mathbf{R}(\Delta\theta, \Delta\varphi) \begin{bmatrix} 0 \\ 0 \\ d \end{bmatrix} \right\| = \frac{f_x d \sin(\psi)}{d \cos(\psi)} = f_x \tan(\psi) \quad (\text{R4})$$

If we assume that robot's head is parallel to the reference path (i.e. centerline) and the curve of centerline is small enough, the 3D position error e_{pos} can be approximated as

$$e_{pos} \approx d_a \tan(\psi) \quad (\text{R5})$$

where d_a is a fixed length along the centerline from the nearest waypoint to a far waypoint P_a pointed by \hat{v} , as shown in **Fig. R1a**. Thus, an approximate relation between e_{img} and e_{pos} can be obtained as

$$e_{img} = \frac{f_x}{d_a} d_a \tan(\psi) \approx \frac{f_x}{d_a} e_{pos} \propto e_{pos} \quad (\text{R6})$$

According to Eq. R6, it can be observed that e_{pos} has positive relation with e_{img} . However, in realistic bronchoscopy procedures, the two hypotheses (i.e., robot is parallel to airway centerline, and the centerline has small curve) are not always satisfied. Thus, we calibrate the pixel-to-millimeter conversion ratio $\Delta e = e_{pos}/e_{img}$ in a statistical way, generating a look-up figure as a reference which records Δe of every position in bronchus.

Calibration of pixel-to-millimeter conversion ratio: We randomly sample 13620 positions along four reference paths of two airway models in the virtual environment, as shown in **Fig. R1c** and **d**. For each sample, the position and direction of robot head is randomly posed around the reference path and the image error e_{img} is calculated by policy network prediction and camera projection as the above process. The position error e_{pos} is measured by the Euclidean distance between robot's head and the reference path, which can be easily accessed in virtual environment. Then the pixel-to-millimeter conversion ratio $\Delta e = e_{pos}/e_{img}$ is calculated for each sample. The statistical results of Δe are shown in **Fig. R1b**, and the mean results at every position of reference path are overlaid on Phantom 1 and 2, as displayed in **Fig. R1e, g** and **Fig. R1f, h**, demonstrating a decrease of Δe with the increase of bronchial generation. It's reasonable for these results because with the bronchoscope gradually going deeper, the bronchus becomes narrower and the e_{pos} becomes smaller, while e_{img} is only determined by robot's direction, so that Δe is smaller. Thus, according to **Fig. R1b**, Δe is 0.075mm/pixel in trachea (0th generation) and 0.018mm/pixel in 9th generation of bronchi, thus, 50-pixel image error means approximate 3.75mm and 0.91mm in 0th and 9th generation of bronchi, respectively. By averaging all samples in **Fig. R1b**, the mean Δe is 0.064mm/pixel. In our in-vivo animal experiment, AI co-pilot group has a mean image error of 11.38 pixels, which reflects a mean 3D position error of 0.73mm in whole procedures.

In addition, the above contents have been added in the revised manuscript. Please check Section **In-vivo Demonstration with Live Porcine Lung Model**, and **Supplementary Note 9** for details.

Comment 4:

What is actuation displacement? What are the primary outcomes as measured in the study?

Response to Comment 4:

We appreciate the reviewer's thoughtful inquiries. Actuation displacement refers to the displacement of the linear motor for pulling the tendon to control the steering of the bronchoscope catheter, as depicted in **Fig. R2a**. To some extent, it can reflect the performance of the control algorithm. In this paper, we recorded the actuation displacements and plotted them. **Fig. R2b** (**Fig. 5c** in the revised manuscript) reflects the actuation displacement of in vivo experiments. It is observed from the heatmap that the AI-human shared control has smaller mean value and fluctuation range than the expert's teleoperation, indicating the higher stability and less jitter.

Fig. R2 a, Internal structure of steering control system used for steer the bronchoscope catheter. **b**, Actuation displacement curves and heatmap of the mean value and fluctuation range of in vivo experiments.

Comment 5:

The introduction suggests that this technology might be useful in “under-privileged” or resource-limited areas where qualified providers are not available. However, the system and technology seem like they would be costly. Is this practical for such resource-limited institutions to acquire? If not, this technology could still have value for providing technical improvement even for experienced users.

Response to Comment 5:

We truly appreciate the reviewer’s insightful suggestion on robot’s cost and the positive comment on **the value of our robot system for providing technical improvement even for experienced users**. To demonstrate the potential of the robot used in “under-privileged” or resource-limited areas, we counted the robotic system's composition and calculated the total cost as depicted in **Supplementary Table 4** in the revised manuscript. The total cost of our system is **\$4168**, which can be afforded by most healthcare institutions, thereby making our technology accessible on a broader scale. Even we retrofit four advanced high-precision force sensors (QLA414, FUTEK, California, America) with a capacity of 22N and a resolution of $\pm 0.5\%$ in our system to monitor actuation force during the bronchoscopy, the total cost of **\$13476** still remains low compared with Monarch Platform (**\$500,000**, reported by <https://www.therobotreport.com/auris-health-220m-monarch-medical-robot/>) and Ion Endoluminal System (**\$204,000 per year**, reported by <https://www.highergov.com/contract-opportunity/6515-lease-ion-endoluminal-robotic-endoscopy-syst-36c24623q0870-s-d95ec/>).

Additionally, we present an alternative force sensor option, the LM15 sensor from Shenzhen Lichi Sensing Technology Co., LTD, with a capacity of 20N and a resolution of 0.1N. Priced at **\$137** per unit, this alternative retains the crucial functionality of force

sensing while significantly reducing the overall cost. The cumulative effect of this substitution brings the total cost of the robotic system to approximately **\$4716**.

Besides, our proposed system can also reduce labor costs, enabling novice doctors to perform bronchoscopy as competently and safely as experienced specialist, reducing the learning curve and ensuring consistent quality of care for equalizing the distribution of medical resources. And even for experienced specialist, our AI co-pilot robot system could still have value for providing technical improvement in bronchoscopy procedures.

revised content:

Supplementary Table 4. Robot costing. The standard version of the system without force sensors has the cost of \$4168, while the advanced version of the system with force sensors has the cost of \$13476.

Equipment or mechanical parts	Model	unit price	number	total price	remarks
linear motor	LA-50	\$273	4	\$1,092	used for steering the catheter
electric slide	EZSM3E040AZMK	\$861	1	\$861	used for feeding the catheter
slide drive module	AZD-KD	\$350	1	\$350	used for driving the electric slide
micro camera	OCHTA10	\$525	1	\$525	used for endoscopic imaging
snake tube	—	\$410	1	\$410	used to make the distal section
braided mesh tube	—	\$410	1	\$410	used to make the proximal section
waterproof rubber	—	\$110	1	\$110	used as a waterproof layer
machined parts	—	\$410	1	\$410	used as connecting parts or shells
force sensor*	qla414	\$2,327	4	\$9,308	used for force limit
Standard version price		\$4,168	Advanced version price		\$13,476

"*" means optional. The device be eliminated or substituted.

Comment 6:

Line 253 suggests that the AI integration “greatly reduces the doctor’s physical exertion and cognitive load during bronchoscopy operation.” This statement does not appear to be well supported by the data. Primarily this data is from Figure 5g which demonstrates a reduced “intervention ratio”. Is there evidence that this is directly related to physical fatigue or cognitive strain? Usually these can be measured with other tools such as the NASA-TLX survey.

Response to Comment 6:

We appreciate the reviewer's insightful suggestion regarding our work and apologize for not providing more evidence that is directly related to physical fatigue and cognitive strain in the previous version of the manuscript. We have added NASA Task Load Index (TLX) survey to this work for validation. An attending doctor who has a little experience in bronchoscopy and a chief doctor who has a lot of experience in both hand-held and robotic bronchoscopy are invited to this survey. Each doctor is asked to perform three trials of robotic bronchoscopy in two bronchial phantoms using our robot system with and without AI-copilot respectively. After every attempt at bronchoscopy,

doctors are requested to complete a NASA TLX questionnaire. The comparison results between the operation with and without AI co-pilot are depicted in **Fig. R3 (Supplementary Fig. 20** in the revised manuscript). It demonstrates that the AI co-pilot operation can significantly reduce the physical fatigue and cognitive strain of users.

Fig. R3 Multi-dimensional reports with and without AI co-pilot acquired by NASA-TLX questionnaires. It contains six subjective subscales, i.e., mental demand (MD), physical demand (PD), temporal demand (TD), performance (P), effort (E) and frustration (F). The smaller rating represents less user task load. The error bar denotes the 95% confidence intervals.

Specifically, the NASA TLX evaluates the perceived workload of humans across six subjective subscales. These subscales include mental demand (the level of mental exertion required by the task), physical demand (the level of physical effort required by the task), temporal demand (the sense of urgency or pace associated with the task), performance (the degree of success achieved in completing the task), effort (the amount of exertion needed to accomplish the performance level), and frustration (the extent of negative emotions such as insecurity, discouragement, irritation, stress, or annoyance experienced during the task). Each subscale is rated on a scale from 0 (very low) to 100 (very high), except for performance, which ranges from 0 (perfect) to 100 (failure).

Please check Section **In-vivo Demonstration with Live Porcine Lung Model** and **Supplementary Note 10** for details.

Comments about the bronchoscopy system:

Comment 7:

It is unclear from the manuscript if this is a guided bronchoscopy platform or if it strictly a mechanized bronchoscopy device with enhanced stabilization. Does it use electromagnetic navigation, CT planning or other features to provide a pathway to the lesion? If so, how does the hardware integrate pre-procedural planning information?

Response to Comment 7:

We appreciate the suggestion regarding our work. The system proposed in this paper is not only a guided bronchoscopy platform or a mechanized bronchoscopy device. It is a robotic bronchoscope platform, consisting of mechanical structures, electric control system and control algorithm for deep lung examination, and has higher intelligence to enable novice doctors to perform bronchoscopy as competently and safely. **Fig. 2** and **Supplementary Note 1 to 3** detailedly describe the composition and control method of the proposed robot.

On the other hand, presently, our robot operates without the use of electromagnetic navigation, CT planning, or similar functionalities to pre-determine pathways to target lesions. The pathway is determined by the operator, with AI policy enhancing intubation control to prevent collisions between the bronchoscope catheter tip and the bronchial walls. Our approach maintains operator decision-making for pathway while leveraging AI for accuracy and safety. However, we envision a future where our system integrates electromagnetic navigation and CT planning, enhancing automation and safety.

Comment 8:

A 1.1 working channel seems quite small. Most tools (needles, forceps, etc) are at least 1.5mm in diameter. What applications do the authors envision the platform being useful for? Strictly lavage?

Response to Comment 8:

We appreciate the reviewer's thorough and insightful comment. Our bronchoscope robot platform can be used for lavage or biopsy operations when equipped with 3.3mm catheter that has a 1.2mm working channel, akin to the 3.8mm hand-held bronchoscope (aScope 3 Slim, Ambu, Baltorpbakken, Denmark). We have performed additional experiment on a live miniature pig to demonstrate the application of the proposed bronchoscope robot. **Fig. R4a** depicts the 3.3mm catheter inserted into a 1mm biopsy forceps, and **Fig. R4b** and **c** shows the endoscopic view of the closed and open biopsy forceps in the miniature pig, respectively.

Fig. R4 a, External view of the 3.3mm catheter inserted into a 1mm biopsy forceps. **b**, Endoscopic view of the closed biopsy forceps. **c**, Endoscopic view of the open biopsy forceps.

Comment 9:

The is the degree of flexion and bend radius of the catheter? From the video it looks like almost 180 degrees but I don't think this is stated in the manuscript.

Response to Comment 9:

We appreciate the reviewer's careful examination and valuable comment. To illustrate the degree of flexion and bend radius of the catheter, we respectively steer the two catheters to the maximum bending angle, as depicted in **Fig. R5**. The 3.3mm catheter with 35mm snake bone has a maximum bending angle of 180 degrees and a bending radius of 5.6mm, while the 2.1mm catheter with 25 mm snake bone has a maximum bending angle of 180 degrees and a bending radius of 4.0mm.

Fig. R5 External view of 2.1mm catheter and 3.3mm catheter at maximum bending angle of 180°.

Revised content:

Supplementary Note 1

The proximal length of the two catheters is 650mm, while the snake bone length of the 3.3mm and 2.1mm catheter are respectively 35mm and 25mm. The distal section of the two catheters can achieve an omnidirectional bending of about 180 degrees for deep lung examination.

Comment 10:

Also, it is unclear how the system is controlled. Perhaps some photos or a video of the controller during the procedure would be helpful.

Response to Comment 10:

To demonstrate how the system is controlled, we show more details about robot control. **Fig. R6a** exhibits the operation scenario of bronchoscope robot system. The operator holds the teleoperator to input the commands, and the PC terminal converts it into the actuation displacement of the linear motor, then the low-level controller control the motor to steer the bronchoscope catheter.

Fig. R6 a, The operation scenario of bronchoscope robot system. **b**, Internal structure of steering control system used for steer the bronchoscope catheter. **c**, Catheter states correspond to the handheld teleoperator at eight positions under teleoperation control.

To illustrate the control principle, we have added a teleoperation experiment demo in **Supplementary Movie 1**. Besides, we provide a photo about the internal structure of the steering control system to illustrate the actuation principle, as depicted in **Fig. R6b**. The system employs tendon-driven mechanism, leveraging four linear motors for pulling the tendon to precisely steer the bronchoscope catheter. **Fig. R6c** shows the teleoperation control process of the bronchoscope robot. We hold the teleoperator at eight's poses (up, down, left, right, up left, down left, up right, down right), then the catheter is controller to the corresponding states. The mapping relationship between the workspace of teleoperator and the configuration parameters of the catheter can be represented as follows.

$$\theta = \theta_{\max} \sqrt{x_t^2 + y_t^2} \quad (\text{R6})$$

$$\varphi = \tan^{-1} \left(\frac{y_t}{x_t} \right) \quad (\text{R7})$$

where (x_t, y_t) is the motion point on the XY plane of the workspace of the teleoperator. θ_{\max} is the preset maximum bending angle, and θ and φ are the configuration parameters of the catheter.

Comments regarding Figures:

Comment 11:

Figure 2 is very dense and hard to follow.

Response to Comment 11:

We apologize for providing confusion in **Fig. 2** of our previous manuscript. We have revised **Fig. 2** in the revised manuscript in a more modular and clearer way to clarify our pipeline.

Overall, **Fig. 2** depicts the working pipeline and the training strategy of our AI-human shared control algorithm. Specifically, **Fig. 2a** illustrates the workflow of the algorithm. The algorithm's core is a policy network that takes a bronchoscopic image and a discrete human command (up, down, left, right, or forward) as input, predicting a steering action (pitch and yaw angle rate) for the robot's orientation, which can be converted to tendon actuation by inverse kinetics and low-level controller.

As shown in **Fig. 2b**, the policy network training process consists of three steps: (a) virtual bronchoscopy environment establishment; (b) data preparation; (c) Sim2Real adaptation. In the first step, an airway model is segmented from the pre-operative CT volume to establish a virtual bronchoscopy environment. The airway centerlines are extracted by the Vascular Modelling Toolkit (VMTK) as reference paths. By simulating a bronchoscope robot in the virtual environment, we can render its observed image and depth. In the second step, the human command and action supervision for each image are automatically generated by an artificial expert agent (**Fig. 2c**) guided by the privileged robot pose and reference airway centerlines, resulting in a training sample, i.e., image, depth, human command and steering action. In the third step, we propose a Sim2Real adaptation module (**Fig. 2d**) to enhance the diversity and photorealism of training samples. The domain adaptation part translates rendered images to a more realistic style while preserving bronchial structure using depth supervision, ensuring the corresponding action supervision remains invariant. The domain randomization part randomly alters image appearances or adds noise to human commands. Upon the dataset prepared above, the data aggregation algorithm (Dagger) is employed for on-policy artificial expert imitation to eliminate distribution mismatch. As every training sample is generated automatically, the entire training process is intervention-free.

revised content, on page 5 of the main text:

Results

AI Co-pilot Bronchoscope Robot Design

Fig. 2a illustrates the overview workflow of the algorithm. The algorithm's core is a policy network that takes a bronchoscopic image and a discrete human command (up, down, left, right, or forward) as input, predicting a steering action (pitch and yaw angle rate) for the robot's orientation, which can be converted to tendon actuation by inverse kinetics and low-level controller. As shown in Fig. 2b, the policy network training process consists of three steps: (a) virtual bronchoscopy environment establishment; (b) data preparation; (c) Sim2Real adaptation. In the first step, an airway model is segmented from the pre-operative CT volume to establish a virtual bronchoscopy environment. The airway centerlines are extracted by the Vascular Modelling Toolkit (VMTK) as reference paths. By simulating a bronchoscope robot in the virtual

environment, we can render its observed image and depth. Supplementary Note 4 presents the virtual environment and simulated robot configurations. In the second step, the human command and action supervision for each image are automatically generated by an artificial expert agent (AEA) guided by the privileged robot pose and reference airway centerlines, resulting in a training sample, i.e., image, depth, human command and steering action. In the third step, we propose a Sim2Real adaptation module to enhance the diversity and photorealism of training samples. The domain adaptation part translates rendered images to a more realistic style while preserving bronchial structure using depth supervision, ensuring the corresponding action supervision remains invariant. The domain randomization part randomly alters image appearances or adds noise to human commands.

Fig. 2 AI-human shared control algorithm and training strategy. **a**, Overview of AI-human shared control algorithm. The bronchoscope robot takes bronchoscopic image during procedure, and doctor gives discrete human command (e.g. left, down, right, up or forward) to decide the direction of robot. Bronchoscopic image and human command are inputted in to a trained policy network to predict continuous steering actions to control the catheter tip centered at the bronchial lumen for safety. Predicted steering actions are then converted to control quantities of linear motors by inverse kinetics and low-level controller, forming a closed loop control system. **b**, Policy network and training strategy. The policy network is designed as a multi-task structure, where the main task is steering action prediction and the side task is depth estimation. Firstly, an airway model is segmented from the pre-operative CT volume to establish a virtual environment. The airway centerlines are extracted as reference paths. Bronchoscopic image and depth are observed by a simulated robot within the virtual environment through rendering. Secondly, an artificial expert agent (AEA) automatically generates human commands and action supervision, producing a training sample containing the image, depth, human command, and steering action. Thirdly, the Sim2Real adaptation module is proposed to enhance the diversity and photorealism of training samples. **c**, Artificial expert agent. This agent has priority to access information of robot and environment for annotation in training process. Two points P_a and P_f on centerline are selected as decision points according to the relative position of robot to the centerline. P_a is used to determine the ground truth steering action by calculate the relative rotation angle to current robot's posture. P_f is farther than P_a and used to determine the human command by projecting the point into the image coordinate system of robot, which is intuitive that human commands usually reflect an expectation that doctor wants robot to reach in the near future. **d**, Sim2Real adaptation module. This module firstly translates rendered images to a more realistic style while preserving bronchial structure. Then four techniques are employed to enhance the generalization capability of the policy network: (i) random rotation of the airway model's roll angle; (ii) random adjustment of the bronchoscope's light intensity; (iii) random noise addition to human commands when the distance between the robot and bronchial wall is $<1\text{mm}$; (iv) random alteration of input image's brightness, contrast, saturation, and hue.

Comment 12:

Figure 3 – Figure labels could be improved. For instance, how does figure 3c differ from 3d? In the text it sounds as though 3d is related to path length whereas 3C is related to simply correct vs. incorrect path. This should be clear from the figure.

Response to Comment 12:

We appreciate the reviewer's careful examination and valuable comment. **Fig. 3c** represents the success rate, i.e. the ratio of successful paths to all paths, showing the generalization ability of each method to reach different branches of the bronchial tree. **Fig. 3d** represents successful path ratio, i.e. the completed path length over the total path length of every single path, showing the coverage ability of each method in the whole bronchial tree. To clarify the criteria used in simulation experiment, we have

revised figure labels of **Fig. 3c** and **Fig. 3d** as “Successful paths / All paths” and “Completed path length / Total length”, respectively.

Besides, we have added definitions of reference path, successful path, success rate and successful path ratio in the revised manuscript for better understanding. A reference path is defined as a centerline of the airway model from the start point of trachea to the end point of terminal bronchus. For example, a total of 60 reference paths are extracted in our testing airway model, shown as green lines in **Fig. R7a**. A successful path is defined as follows: the simulated robot reaches the range within 1cm of the end point of the reference path, without any collision and wrong path choice during the running process, shown as the red line in **Fig. R7b**. A collision is occurred if the distance between the robot and the inner wall is less than 0.1mm, as shown in **Fig. R7c**. A wrong path choice is detected if the robot exceeds a virtual tunnel around the reference path, as shown in **Fig. R7d**.

Fig. R7 (Supplementary Fig. 18) Reference paths and criteria of successful path. **a**, Reference paths (i.e., centerlines) of Patient 3 for testing. **b**, An example of successful path that simulated robot reaches within a range of 1cm from the end point of the reference path. **c**, An example of failed path where the collision occurs between simulated robot and the inner bronchial wall. **d**, An example of failed path where the simulated robot enters into a wrong lumen.

Also, we have added definitions of success rate (SR) and successful path ratio (SPR). The success rate is defined as the ratio of successful paths to all paths:

$$SR = N_s/N_{all}$$

For each reference path, the successful path ratio is defined as the ratio of completed length to the total length of the reference path:

$$SPR_i = L_s^i / L_{all}^i$$

where L_s^i is the path length along reference path from the start point to the stopping point of simulated robot. If the robot successfully reaches the end point, the SPR_i will be set to 1. If robot fails in the process, the nearest waypoint on the reference path is regarded as the stopping point to calculate L_s^i .

Please check **Fig. 3** and **Supplementary Note 9** for details.

revised content:

Fig. 3 Results of the simulation experiments. a, An example of an airway model containing 5th-generation bronchi with a reference path for bronchoscopy. **b**, Two styles of images rendered from airway models with different texture mapping. Sim-

style images use a pink texture, while real-style images employ a realistic texture. **c**, Success rates (i.e. the ratio of successful paths to all paths, showing the generalization ability to reach different branches of the bronchial tree for each method). **d**, Successful path ratios (i.e. completed path length over the total path length of every path, showing the coverage ability in the whole bronchial tree for each method). **e**, Trajectory errors of different methods running in the testing environment with realistic texture, containing 60 paths for evaluation. **f**, Qualitative image translation results of different methods, where Sim represents the Sim-style images as the source domain, and Real, Phantom, and Clinical denote three styles of realistic images as target domains. The training datasets of the source and target domain are unpaired. AttentionGAN is chosen as the baseline method for unpaired image translation. Detailed illustrations of training datasets and image translation results are depicted in Supplementary Fig. 8. **g**, Structural similarity index measure. **h**, peak signal-to-noise ratio results of different methods, where higher SSIM and PSNR values indicate better structure-preserving properties.

Comment 13:

Figure 4 – for d and e, what is depicted in the bar plots? Is it average image error over the duration of the procedure?

Response to Comment 13:

We appreciate the review’s insightful comment. The bar plot in **Fig. 4d** depicts the average image error with 95% confidence interval (CI) in Phantom 1, including two bronchoscopy procedures along Path 19 and Path 55 respectively for each group (e.g. AI and Expert). In each group, all frames during the two bronchoscopy procedures are counted for calculating average image error with 95% CI and creating the bar plot. Similarly, the bar plot in **Fig. 4e** depicts the average image error with 95% CI in Phantom 2, including Path 25 and Path 66.

For a clearer presentation of meaning of bar plots, we have revised the caption of **Fig. 4** in the revised manuscript.

revised content:

Fig.4 Results of in-vitro experiments. **a**, In vitro experimental scenario under the breathing simulation, with detailed design and motion analysis illustrated in Supplementary Fig. 10. **b**, The airway models correspond to the two bronchial phantoms. **c**, Learning process of the novice doctors. Path 23 in Phantom 1 is selected for bronchoscopy. Two novice doctors (a medical intern and a attending doctor) are invited to learn teleoperated bronchoscopy with and without AI co-pilot respectively from demonstrations. The medical intern performs three bronchoscopy trials without AI co-pilot. Then, the expert without AI co-pilot and the attending doctor with AI co-pilot perform bronchoscopy, separately. Image errors during the procedure are recorded and statistically analyzed. **d**, Comparison results in Phantom 1. Path 19 and 55 are selected for bronchoscopy, covering both sides of Phantom 1. The expert without AI co-pilot and the attending doctor with AI co-pilot are required to perform bronchoscopy separately. **e**, Comparison results in Phantom 2. Path 25 and 66 are selected for bronchoscopy. The results reveal that the novice doctors with AI co-pilot can achieve and maintain a smaller image error than the expert during the bronchoscopy procedure, even when the initial position of the AI co-pilot group is less favorable, as in the case

of Phantom 1 Path 66. Bar plots depict the mean image error with 95% confidence interval (CI) of all frames during two bronchoscopy procedures in each phantom, i.e. mean \pm 95% CI image error in Phantom1 containing Path 19 and Path 55, and Phantom 2 containing Path 25 and Path 66.

Comment 14:

Figure 5 – in c and d, which actuation displacement curve corresponds to which condition? Is it the same order as 5b?

Response to Comment 14:

The actuation displacement curve in **Fig. 5c** and **d** corresponds to the trials in **Fig. 5b**. To increase readability, we have marked the condition corresponding to every curve in **Fig. 5 c** and **d** in the revised manuscript.

revised content:

Fig. 5 Results of in-vivo experiments. a, CT images of the live porcine lung. **b,** Bronchoscopic images obtained by the expert without AI co-pilot and the attending

doctor with AI co-pilot correspond to the two paths in Supplementary Fig. 26, respectively. **c**, Actuation displacement curves and heatmap of the mean value and fluctuation range. **d**, Actuation force curves and heatmap of the mean value and fluctuation range. **e**, Image error during bronchoscopy. The results show that the attending doctor with AI co-pilot consistently maintains more minor image errors than the expert without AI co-pilot. **f**, Statistical results of image error. Bar plots depict the mean image error with 95% confidence interval (CI) of all frames during bronchoscopy procedures in live porcine lung, i.e. mean \pm 95% CI image error of Path 1 and Path 2. The results show the safer steering of AI-assisted bronchoscopy. **g**, Statistical results of the human intervention ratios, determined by the frequency of human control adjustments relative to the total duration of recorded data in bronchoscopy procedures. The results demonstrate that AI-assisted bronchoscopy can effectively minimize the physical strain and cognitive burden on doctors.

Comment 15:

Supplemental Figure 11 – for a and b, the figure legend mentions 5 groups but to my eye there are only 4 sets of curves. This should be clarified.

Response to Comment 15:

We apologize for missing a curve in **Supplemental Fig. 11** in the previous manuscript. We have revised this figure and marked the experimental conditions corresponding to each curve to improve readability in the new version of manuscript.

revised content:

Supplementary Fig. 14 Actuation information on a specified path obtained by the participants. **a**, Actuation displacement curves obtained by the medical intern (the first three groups), the expert (the fourth group), and the attending doctor with AI co-pilot (the fifth group). **b**, Actuation force curves obtained by the medical intern (the first three groups), the expert (the fourth group), and the attending doctor with AI co-pilot (the fifth group). **c**, Distribution of actuation displacements. **d**, Distribution of actuation forces.

Comment 16:

Supplemental Figure 14 – a through d. These conditions need to be better labelled. 14a for instance talks about data from experiments in 2 phantoms from 1 user but there are 4 sets of 4 curves.

Response to Comment 16:

We appreciate the reviewer’s careful reading and kind suggestion. We apologize for confusingly showing the curves in **Supplemental Fig. 14**. We have revised this figure to improve readability.

revised content:

Supplementary Fig. 16 Actuation information on multiple paths obtained by the participants. a, Actuation displacement curves obtained by the expert with teleoperation on two phantoms. **b,** Actuation force curves obtained by the expert on two phantoms. **c,** Actuation displacement curves obtained by the attending doctor with AI co-pilot on two phantoms. **d,** Actuation force curves obtained by the attending doctor with AI co-pilot on two phantoms. **e,** Distribution of actuation displacements obtained by the expert and the attending doctor with AI co-pilot. **f,** Distribution of actuation forces obtained by the expert and the attending doctor with AI co-pilot.

Comment 17:

Movies – The movies are interesting however several of the videos are very blurry and the zip files takes over an hour to download.

Response to Comment 17:

We thank the reviewers for pointing out this important movie issue. Due to the size limitation of uploaded files, we have compressed the videos and submitted them. Besides, we also uploaded it to these website:

https://drive.google.com/drive/folders/1IaUV3hVNP3DbTc3a80sjwREsPr2MouUH?usp=share_link

<https://www.aliyundrive.com/s/qo7rfEi3yNE>

We hope these high definition videos are now easy to fetch.

Reviewer #2 (Remarks to the Author):

General Comment:

This work proposes an AI policy network for guiding a robot during bronchoscopy. The AI shares control with a surgeon so provide safer motions when navigating through the lungs. The authors' tested their robot in both in-vitro and in-vivo settings with different levels of experienced surgeons. This paper would be of interest in the field of surgical robotics and machine learning. The focus is mainly on the implementation of an AI software as a method for sharing control of the robot with a human. The methods used by the authors result in a novel contribution to the field. A few questions and considerations remain when reading this work that would help strengthen and clarify the conclusions.

Response to General Comment:

We are grateful for your careful review and valuable suggestions, which have enabled us to greatly improve our article. We also feel encouraged that our work would be of interest in the field of surgical robotics and machine learning, and also results in a novel contribution to the field. Following the your insightful comments and kind suggestions, we have revised our manuscript with the most seriousness and please find our detailed responses below.

Comment 1:

In the introduction, the authors make valid claims about the potential issues that may arise in traditional bronchoscopy. Adding some references, if available, to quantify the frequency or severity of these issues would help better frame the proposed robot with respect how the procedure is currently performed.

Response to Comment 1:

Thanks for the reviewer's kind suggestions, we have supplemented the references [19-21] in the revised manuscript to quantify the frequency or severity of the bronchoscope robots in disease diagnosis.

revised content, on page 9 of the main text:

Introduction (underlined text indicates added content)

The Monarch Platform is equipped with an internal bronchoscope catheter with 4.2 mm diameter, and an external sheath of 6 mm. Its subtle steering control and flexibility, allowing for deeper access into the peripheral regions of the lungs, surpassing conventional bronchoscopes¹⁷ (9th vs. 6th airway generations). In parallel, the Ion Endoluminal System boasts a fully articulated 3.5mm catheter with a 2mm working channel, enhanced stability, superior flexibility, and the added advantage of shape perception¹⁸. Notably, studies indicate that these platforms exhibit a favorable diagnostic yield, ranging from 81.7% to 92%, for lung nodules with sizes between 14.8

mm and 21.9 mm¹⁹⁻²¹. Moreover, the complication rates reported are minimal. These findings suggest that these platforms play a transformative role in the future management of pulmonary conditions. In addition to the Monarch and Ion platforms, several other bronchoscope robotic systems developed by academic institutions are under development or in early-stage research to address sensing and control issues for doctors²¹⁻²⁶. Despite its advantages, current telerobotic bronchoscopy faces several challenges, including the learning curve and lack of autonomy.

[19] Benn, B. S., Romero, A. O., Lum, M. & Krishna, G. Robotic-assisted navigation bronchoscopy as a paradigm shift in peripheral lung access. *Lung* **199**, 177-186 (2021).

[20] Kalchier-Dekel, O. et al. Shape-sensing robotic-assisted bronchoscopy in the diagnosis of pulmonary parenchymal lesions. *Chest* **161**, 572-582 (2022).

[21] Ost, D. et al. Prospective multicenter analysis of shape-sensing robotic-assisted bronchoscopy for the biopsy of pulmonary nodules: results from the PRECISE study. *Chest* **160**, A2531-A2533 (2021).

Comment 2:

While few state-of-the-art robot bronchoscopes are mentioned, there should be more detail to explain their relevance with respect to the proposed work. Also, there are more state-of-the-art bronchoscopy robots being created in research labs that have shown new tools for bronchoscopy, implementations of control, and safe navigation. Adding greater detail and more references to the state-of-the-art would help better frame the authors' work. The authors can see this list of references as a starting point for bronchoscopy robots that are currently being researched:

1. Bao, Y., Li, X., Wei, W., Liu, H., Qu, S. (2022). Study on the interventional path planning method of bronchoscope catheter. Journal of Mechanical Science and Technology 2022 36:5, 36(5), 2643–2652.

2. Van Lewen, D., Janke, T., Lee, H., Austin, R., Billatos, E., Russo, S. (2023). A Millimeter-Scale Soft Robot for Tissue Biopsy Procedures. Advanced Intelligent Systems, 2200326.

3. Pittiglio, G., Lloyd, P., da Veiga, T., Onaizah, O., Pompili, C., Chandler, J.H. and Valdastrì, P., 2022. Patient-specific magnetic catheters for atraumatic autonomous endoscopy. Soft Robotics, 9(6), pp.1120- 1133.

4. McCandless, M., Perry, A., DiFilippo, N., Carroll, A., Billatos, E., Russo, S. (2022). A Soft Robot for Peripheral Lung Cancer Diagnosis and Therapy. Soft Robotics, 9(4), 754–766.

Response to Comment 2:

Thanks for the reviewer's insightful comment, we have supplemented the recommended references [22-24] and [33] in the revised manuscript to improve the readability and persuasiveness.

revised content:

The Monarch Platform is equipped with an internal bronchoscope catheter with 4.2 mm diameter, and an external sheath of 6 mm. Its subtle steering control and flexibility, allowing for deeper access into the peripheral regions of the lungs, surpassing conventional bronchoscopes¹⁷ (9th vs. 6th airway generations). In parallel, the Ion Endoluminal System boasts a fully articulated 3.5mm catheter with a 2mm working channel, enhanced stability, superior flexibility, and the added advantage of shape perception¹⁸. Notably, studies indicate that these platforms exhibit a favorable diagnostic yield, ranging from 81.7% to 92%, for lung nodules with sizes between 14.8 mm and 21.9 mm. Moreover, the complication rates reported are minimal¹⁹⁻²¹. These findings suggest that these platforms play a transformative role in the future management of pulmonary conditions. In addition to the Monarch and Ion platforms, several other bronchoscope robotic systems developed by academic institutions are under development or in early-stage research to address sensing and control issues for doctors²²⁻²⁷. Despite its advantages, current telerobotic bronchoscopy faces several challenges, including the learning curve and lack of autonomy.

Integrating artificial intelligence (AI) techniques into bronchoscopy further expands the horizons of this burgeoning field²⁸. By leveraging advanced algorithms, such as machine learning and computer vision technologies²⁹, researchers are developing image-guided navigation systems, process and interpreting bronchoscopic imagery³⁰, facilitating the real-time localization³¹, tracking³² and interventional path planning³³ of endoscopy, and enabling precise navigation within the bronchial tree. These software systems enhance the accuracy and efficiency of bronchoscopy procedures. Furthermore, by providing automated, continuous guidance throughout the procedure³⁴, image-guided systems can help to reduce the cognitive load on the operating doctor, allowing them to focus on other critical aspects of the procedure³⁵. However, these systems present safety concerns during bronchoscopy procedures, as they rely on bronchoscope localization in pre-operative CT³⁶⁻³⁸, which may suffer from misregistration and unsafe steering for robots due to limited field of view and body-CT visual discrepancies. Concerns about the risk of complications, such as pneumothorax and bleeding, have been raised, underlining the need for ongoing research and optimization of these platforms.

[22] Van Lewen, D. et al. A Millimeter-Scale Soft Robot for Tissue Biopsy Procedures. *Adv. Intell. Syst.* **5**, 2200326 (2023).

[23] Pittiglio, G. et al. Patient-specific magnetic catheters for atraumatic autonomous endoscopy. *Soft Robot.* **9**, 1120-1133 (2022).

[24] McCandless, M. et al. A soft robot for peripheral lung cancer diagnosis and therapy. *Soft Robot.* **9**, 754-766 (2022).

[33] Bao, Y. et al. Study on the interventional path planning method of bronchoscope catheter. *J Mech Sci Technol.* **36**, 2643–2652 (2022).

Comment 3:

The idea of centering the bronchoscope within the lung branch during navigation will benefit the safety of the patient. In general, navigating a bronchoscope requires some interaction with the lung walls when making turns. The robot may not be able to stay centered along the full length it has traveled. Will the task of staying centered at the distal tip concentrate extra stress on the lung walls elsewhere along the robot's body?

Response to Comment 3:

We truly appreciate the good question offered by the reviewer, as this perspective holds immense research value and significance. Due to the passive compliance of proximal section, staying centered at the catheter tip cannot avoid concentrating extra stress on the lung walls. To assess the magnitude of the contact force, we supplement a contact force measurement experiment during the movement of the bronchoscope catheter, as depicted in **Fig. R8a**. Three pipes were placed on the table and the bronchoscope catheter was inserted to simulate the contact circumstances with the bronchial walls. The uniaxial force sensor is installed under each pipe to measure the contact force. The catheter tip is controlled to steer in the vertical plane, and the measurement data of each sensor is recorded and plotted. It is observed from **Fig. R8b** that as the bending angle increases in both directions, the contact force between the catheter and the pipe walls increases. However, the force is relatively small, maintaining within 0.3N. In contrast, the collision of the catheter tip may cause the lung walls to rupture or bleed. This demonstrates a critical inference — averting collisions between the catheter tip and the airway wall stands as the most pivotal consideration. Presently, the primary focus of intubation performed by the expert doctors revolves around staying centered at the catheter tip to avoid collision of the catheter tip. And, this objective aligns closely with the central focus of our algorithm optimization efforts.

Fig. R8 a, The experimental scene. **b**, The contact force variation measured by the force sensor when the catheter is controlled to steer.

Comment 4:

It is unclear if the AI co-pilot is only predicting the steering actions and displaying visual feedback to the surgeon or if it is also applying some intervention and modifying the steering actions of the surgeons. If the latter is the case, would the surgeon be able to take control of the robot from the AI co-pilot if they wanted?

Response to Comment 4:

We apologize for not providing clear explanation about the mechanism of our AI co-pilot system. In our system design, AI co-pilot not only predicting the steering actions and displaying visual feedback to the surgeon, but also applying intervention and modifying the steering actions of surgeons. Specifically, there are two modes in our AI co-pilot system, i.e. teleoperation mode and AI shared control mode, as shown in **Fig. R9 (Supplementary Fig. 3c** in the revised manuscript). In teleoperation mode, the surgeon's command in hand (motion trajectory of the teleoperator) is directly mapped into actuation state to control the omnidirectional bending of the bronchoscope catheter. Differently, in the AI shared control mode, the surgeon inputs the discrete commands (up, down, left, right or forward direction) into the policy network along with the bronchoscopic images, then the steering actions are predicted by AI policy to safely control the bending of the bronchoscope catheter. Besides, the predicted steering actions are projected and displayed in the bronchoscopic images for visualization. Thus, in this mode, our AI co-pilot system not only predicting the steering actions and displaying visual feedback to the surgeon, but also applying interventions.

Fig. R9 The control block diagram of the AI co-pilot bronchoscope robot.

For more intuitive explanation, we have conducted an experiment to demonstrate the switch between two modes. As shown in **Fig. R10 (Supplementary Fig. 21)**, at first, we put the catheter tip close to the bronchial wall using teleoperation mode and keep doctor's hand still for no external control on robot. Then we start the AI shared control mode and doctor's hand also stay still. It's obvious that the image error decreases rapidly and automatically as the AI shared control mode is triggered. It demonstrates that AI co-pilot automatically applies intervention to doctor's command for safe control.

Fig. R10 Control switching experiment between teleoperation mode and AI shared control mode.

In practical use, doctors can take over full control of the robot at any time from the AI co-pilot if they deem it necessary. We have designed an interface to take control in our software-controlled GUI interface, as depicted in **Fig. R11 (Supplementary Fig. 22)**. If the **AI stop button** (marked in a red box) is pressed, the AI shared control mode is disabled, and instead, teleoperation mode is activated, which means the doctor can operate the robot by teleoperation.

Fig. R11 Software-controlled GUI interface.

In addition, the above contents are added to **Supplementary Note 11**, please check them for details.

Comment 5:

In the simulation experiments, the success rate is defined as ratio of successful paths to all paths. The criteria for a successful path was not explained. Does this refer to mapping reference paths to the simulated bronchoscope images? If this is the case, how is the simulated path derived from the simulated bronchoscope images? The same issue is found for the successful path ratio. What are the criteria for a path to

be completed? Further, explanation of the simulation setup and success criteria would help in understanding.

Response to Comment 5:

We apologize for not providing more explanation of simulation setup and definition of success criteria in the previous version of the manuscript. In the simulation experiment, a reference path is defined as a centerline of the airway model from the start point of trachea to the end point of terminal bronchus. For example, a total of 60 reference paths are extracted in our testing airway model, shown as green lines in **Fig. R12a**. A successful path is defined as follows: the simulated robot reaches the range within 1cm of the end point of the reference path, without any collision and wrong path choice during the running process, shown as the red line in **Fig. R12b**. A collision is occurred if the distance between the robot and the inner wall is less than 0.1mm, as shown in **Fig. R12c**. A wrong path choice is detected if the robot exceeds a virtual tunnel around the reference path, as shown in **Fig. R12d**.

Fig. R12 (Supplementary Fig. 18) Reference paths and criteria of successful path. **a**, Reference paths (i.e. centerlines) of the testing airway model. **b**, An example of successful path that simulated robot reaches within a range of 1cm from the end point of the reference path. **c**, An example of failed path where the collision occurs between simulated robot and the inner bronchial wall. **d**, An example of failed path where the simulated robot enters into a wrong branch.

Besides, we have added definitions of success rate (SR) and successful path ratio (SPR). The success rate is defined as the ratio of successful paths to all paths:

$$SR = N_s/N_{all}$$

For each reference path, the successful path ratio is defined as the ratio of completed length to the total length of the reference path:

$$SPR_i = L_s^i/L_{all}^i$$

where L_s^i is the path length along reference path from the start point to the stopping point of simulated robot. If the robot successfully reaches the end point, the SPR_i will be set to 1. If robot fails in the process, the nearest waypoint on the reference path is regarded as the stopping point to calculate L_s^i .

The success rate can measure the generalization ability of the policy network to reach different branches of the bronchial tree and the successful path ratio can evaluate the coverage ability in the whole bronchial tree.

In addition, we have supplemented the above contents in the revised manuscript. Several videos of simulation experiments have been also added for clearer demonstration. Please check **Supplementary Note 9** and **Supplementary Movie 1** for details.

Comment 6:

The in-vitro evaluation procedure with the inclusion of breathing motion is well described and is setup well with both novice and experienced surgeons. The results of both in-vitro and in-vivo experiments are reported in the form of image errors (pixels). It is unclear what the image errors translate to a clinical scenario. Is this a distance, in pixels, between the centers of each image? If this is the case, is image error a relevant benchmark when the difference in error between the AI and expert are on the order of 10 pixels? At such a small scale in bronchoscopy, this difference in image error likely translates to less than a millimeter of error depending on the camera specifications. Further details on this would be helpful in understanding these results.

Response to Comment 6:

We appreciate the reviewer's careful reading and insightful comments. In practical terms, 10-pixel image error means the distance is 10 pixels from image center to bronchial lumen center in the image coordinate system, which reflects approximate 0.64mm 3D position error between the robot's head and the airway centerline in 3D space. Besides, in our in-vivo animal experiment, AI co-pilot group has a mean image error of 11.38 pixels during all bronchoscopy procedures, which reflects a mean 3D position error of 0.73mm. In this study, the pixel-to-millimeter conversion ratio is about 0.064mm/pixel. Commonly, the accuracy of bronchoscopy procedure is directly measured by the 3D position error, which is the Euclidean distance from the robot's head to the nearest waypoint on the reference path (i.e. airway centerline) in 3D space. However, in practical dynamic scenarios, due to respiratory movement and deformation of bronchus, 3D position error is hard to measure using existing electromagnetic and visual tracking techniques, which prone to introduce a large system error in ground

truth reference path localization. Thus, we employ the image error as the metric to measure the accuracy of bronchoscopy procedure, which represents the distance in pixel from image center to bronchial lumen center in the image coordinate system and can also support our conclusions. Specific definition of image error and experiments on pixel-to-millimeter calibration are introduced as follows.

Fig. R13 (Supplementary Fig. 19). Image error to 3D position error mapping. a, Definitions of image error and 3D position error. The image error e_{img} is projected by a predicted direction vector \hat{v} by the policy network, which directs to a point lying on the centerline in 3D space. And the 3D position error e_{pos} is defined as the nearest distance between robot's head and the reference path. **b,** Statistical results of the pixel-to-millimeter conversion ratio in different bronchial generations. **c,** Virtual environment established on the airway model of Phantom 1 with its two reference paths Path 19 and 55. **d,** Virtual environment based on Phantom 2 airway model with Path 25

and 66. **e**, Segments of bronchial generations along Path 19 and 55 in Phantom 1. **f**, Segments of bronchial generations along Path 25 and 66 in Phantom 2. **g**, Distribution of pixel-to-millimeter conversion ratio along reference paths in Phantom 1. **h**, Distribution of conversion ratio along reference paths in Phantom 2. It's obvious that with the bronchoscope robot reaching deep bronchus, the conversion ratio becomes smaller, because the diameter of airway tree becomes thinner.

Definition of image error: The accuracy of bronchoscopy procedures is commonly measured by the 3D position error, which is the Euclidean distance between the robot's head and the reference path (i.e. centerline) in 3D space, shown as e_{pos} in **Fig. R13a**. Some existing works have reported position error results (also called absolute tracking error, ATE) of bronchoscopy in dead porcine lung or static phantom by aligning pre-operative reference path with realistic bronchial tree as the ground truth path. The key hypothesis of these works is that the lung is static and rigid when performing bronchoscopy. However, in clinical scenarios, due to the respiratory motion of live lung and the deformation of bronchus, all existing methods fails to accurately align reference paths with realistic bronchial tree, resulting in inaccurate 3D position error measurement, which is especially severe in deep bronchi. To address this issue, in this study, we use image error to measure the accuracy of bronchoscopy procedure, shown as e_{img} in **Fig. R13a**. The image error is a widely used metric in the field of robot visual serving, which measures the distance from image center to the target position in image coordinate system without the need of 3D position. In practical terms, a mapping from e_{img} to e_{pos} can be formed in a statistical way, which we have performed new experiments for discussion in the following part.

The image error e_{img} is calculated by projecting the next direction vector $\hat{\mathbf{v}} \in \mathbb{R}^3$, predicted by the policy network, into the current image coordinate system of robot. In this study, we assume the policy network is well trained and $\hat{\mathbf{v}}$ should point to a waypoint lying on the centerline, satisfying

$$e_{img} = \|\overline{Op}\| = \left\| \frac{1}{\hat{\mathbf{v}}_z} \mathbf{K} \hat{\mathbf{v}}^T \right\| \quad (\text{R1})$$

where \overline{Op} is projected from $\hat{\mathbf{v}}$ in image, $\hat{\mathbf{v}}_z$ is the z-coordinate value of $\hat{\mathbf{v}}$, $\mathbf{K} = \begin{bmatrix} f_x & 0 & 0 \\ 0 & f_y & 0 \end{bmatrix}$ is the known intrinsic matrix of camera, f_x and f_y are intrinsic parameters and satisfy $f_x = f_y$. Actually, $\hat{\mathbf{v}}$ is predicted by the policy network π and derived from the steering action $a = \pi(x, c) = [\Delta\theta, \Delta\varphi]$. According to the formulation of ground truth steering action in **Fig. 2c**, if the policy network is well trained, $\hat{\mathbf{v}}$ will direct to a point P_a on the centerline, and the projected \overline{Op} will direct to the bronchial lumen in image coordinate system. By using steering action, $\hat{\mathbf{v}}$ can be rotated from the current direction vector $\mathbf{v} = [0, 0, d]$ of robot with a fixed depth d as

$$\hat{\mathbf{v}}^T = \mathbf{R}(\Delta\theta, \Delta\varphi)\mathbf{v}^T = \mathbf{R}(\Delta\theta, \Delta\varphi) \begin{bmatrix} 0 \\ 0 \\ d \end{bmatrix} \quad (\text{R2})$$

where $\mathbf{R}(\Delta\theta, \Delta\varphi) \in \mathbb{R}^{3 \times 3}$ is the rotation matrix from \mathbf{v} to $\hat{\mathbf{v}}$, parameterized by the steering action $[\Delta\theta, \Delta\varphi]$ as

$$\mathbf{R}(\Delta\theta, \Delta\varphi) = \begin{bmatrix} \cos(\Delta\theta) & 0 & \sin(\Delta\theta) \\ \sin(\Delta\varphi)\sin(\Delta\theta) & \cos(\Delta\varphi) & -\cos(\Delta\theta)\sin(\Delta\varphi) \\ -\cos(\Delta\varphi)\sin(\Delta\theta) & \sin(\Delta\varphi) & \cos(\Delta\varphi)\cos(\Delta\theta) \end{bmatrix} \quad (\text{R3})$$

For simplicity, the rotation can also be parametrized by an angle $\psi = f(\Delta\theta, \Delta\varphi)$ about an axis of rotation, as shown in **Fig. R13a**. Thus, we can rewrite Eq. R1 as following

$$e_{img} = \left\| \frac{1}{\hat{\mathbf{v}}_z} \mathbf{K} \mathbf{R}(\Delta\theta, \Delta\varphi) \begin{bmatrix} 0 \\ 0 \\ d \end{bmatrix} \right\| = \frac{f_x d \sin(\psi)}{d \cos(\psi)} = f_x \tan(\psi) \quad (\text{R4})$$

If we assume that robot's head is parallel to the reference path (i.e. centerline) and the curve of centerline is small enough, the 3D position error e_{pos} can be approximated as

$$e_{pos} \approx d_a \tan(\psi) \quad (\text{R5})$$

where d_a is a fixed length along the centerline from the nearest waypoint to a far waypoint P_a pointed by $\hat{\mathbf{v}}$, as shown in **Fig. R13a**. Thus, an approximate relation between e_{img} and e_{pos} can be obtained as

$$e_{img} = \frac{f_x}{d_a} d_a \tan(\psi) \approx \frac{f_x}{d_a} e_{pos} \propto e_{pos} \quad (\text{R6})$$

According to Eq. R6, it can be observed that e_{pos} has positive relation with e_{img} . However, in realistic bronchoscopy procedures, the two hypotheses (i.e., robot is parallel to airway centerline, and the centerline has small curve) are not always satisfied. Thus, we calibrate the pixel-to-millimeter conversion ratio $\Delta e = e_{pos}/e_{img}$ in a statistical way, generating a look-up figure as a reference which records Δe of every position in bronchus.

Calibration of pixel-to-millimeter conversion ratio: We randomly sample 13620 positions along four reference paths of two airway models in the virtual environment, as shown in **Fig. R13c** and **d**. For each sample, the position and direction of robot head is randomly posed around the reference path and the image error e_{img} is calculated by policy network prediction and camera projection as the above process. The position error e_{pos} is measured by the Euclidean distance between robot's head and the reference path, which can be easily accessed in virtual environment. Then the pixel-to-millimeter conversion ratio $\Delta e = e_{pos}/e_{img}$ is calculated for each sample. The statistical results of Δe are shown in **Fig. R13b**, and the mean results at every position of reference path are overlaid on Phantom 1 and 2, as displayed in **Fig. R13e, g** and **Fig. R13f, h**, demonstrating a decrease of Δe with the increase of bronchial generation. It's

reasonable for these results because with the bronchoscope gradually going deeper, the bronchus becomes narrower and the e_{pos} becomes smaller, while e_{img} is only determined by robot's direction, so that Δe is smaller. Thus, according to **Fig. R13b**, Δe is 0.075mm/pixel in trachea (0th generation) and 0.018mm/pixel in 9th generation of bronchi, thus, 10-pixel image error means approximate 0.75mm and 0.18mm in 0th and 9th generation of bronchi, respectively. By averaging all samples in **Fig. R13b**, the mean Δe is 0.064mm/pixel. In our in-vivo animal experiment, AI copilot group has a mean image error of 11.38 pixels, which reflects a mean 3D position error of 0.73mm in whole procedures.

In addition, the above contents have been added in the revised manuscript. Please check Section **In-vivo Demonstration with Live Porcine Lung Model**, and **Supplementary Note 9** for details.

Comment 7:

Also, the paths chosen for these evaluations seem to be straight-forward with less steering required. Is the robot able to make tighter bends into the upper portions of the lung?

Response to Comment 7:

Thanks to the insightful questions of the reviewer. To assess the robot's ability to bend into the upper portions of the lung, we supplement an experiment as depicted in **Fig. R14a**. It is observed that the robot could enter partial upper bronchus, but could only reach the entrance of narrower bronchial passages. In addition, we also conducted tests using a 3.8mm commercial bronchoscope (aScope 3 Slim, Ambu, Baltorpbakken, Denmark), as depicted in **Fig. R14b**. The area that the bronchoscope can reach is basically the same as the robot we proposed.

Fig. R14 a, Steering experiments of our proposed robot in the upper portions of the bronchial phantom. **b**, Steering experiments of the commercial bronchoscope in the upper portions of the bronchial phantom.

The main reason that prevents the robot from entering the narrow upper bronchus is related to the design structure of the tendon-driven method. This kind of bronchoscope robots have a high-stiffness proximal section and low-stiffness distal section, and only the distal section is actively steered under the tendon actuation. The bronchoscope catheter is fed deep into the lung by the follow-the-tip motion. It's important to acknowledge that this method presents certain limitations, particularly evident when entering the upper portions of the lung where a steering angle exceeding 150 degrees of the distal section is often necessary. In this circumstance, the heightened stiffness of the proximal section brings great challenge to feeding the catheter. However, we believe that with the advancement of materials science and actuator technology, this problem can be better solved.

On the other hand, the focus of this paper is to propose an AI co-pilot bronchoscope robot system that uses AI strategy to lower the medical barrier and enables novice doctors to perform bronchoscopy as competently and safely as experienced specialists, reducing the learning curve and ensuring consistent quality of care.

Comment 8:

Familiarity with teleoperation can make a difference in surgeon skills. Have the surgeons used a similar teleoperation platform before these trials were performed? If not, would allowing the surgeons time to train on the teleoperation platform change the results? Any details on this would strengthen the comparison to an expert surgeon since an expert surgeon with no experience in teleoperation may lead to results showing greater error than if the surgeon had teleoperation experience.

Response to Comment 8:

We truly appreciate the reviewer's insightful question. The participants in the experiments don't use a similar teleoperation platform before these trials were performed. To assess whether well-trained doctor would improve outcomes, we invited an expert (with much experience) and a medical intern (without experience) to participate all the experiments. The detailed operation level of these doctors is described in **Supplementary Table 3**.

The experimental results are depicted in **Fig. R15 (Fig. 4c** in the revised manuscript). The curves correspond to **Trial 1, Trial 2, Trial 3** are conducted by the medical intern, while the curve correspond to **Expert** is conducted by the expert. It can be seen that the prior training had a certain impact on the improvement of their operation skills. On the other hand, we also invite an attending doctor without experience to participate with AI co-pilot. By the comparison of the expert and the attending doctor, it is found that the

AI co-pilot can improve the operation skills of the novice doctor to the level of the well-trained doctor.

Fig. R15 Comparison between the well-trained doctor (an expert) and the novice doctors (a medical intern and a attending doctor without experience). The medical intern performs three bronchoscopy trials, and the expert performs one trial without AI co-pilot. The attending performs one bronchoscopy trial with AI co-pilot.

revised content:

Supplementary Table 3. The operation experience of participants in the experiments. The medical intern (novice 1), the attending doctor (novice 2) and the expert doctor (expert) both come from the School of Medicine at Zhejiang University, Hangzhou, China.

Participant	Professional title	Entire period of operation	Manual operations	Robot teleoperations
novice 1	medical intern	no	no	2 demonstrations
novice 2	attending doctor	<5 years	<100 cases per year	2 demonstrations
expert	chief doctor	>20 years	>200 cases per year	>100 trials

Comment 9:

Figure 1 shows the bronchoscope robot; however, the robot components are not well explained in the main text. What kind of micro camera and materials are used to make the robot? This would help with reproducibility of the proposed work.

Response to Comment 9:

We truly appreciate the insightful question offered by the reviewer. The more detailed description of the robot's components has been added in **Supplementary Note 1**. Our bronchoscope catheter is installed a micro camera (OCHTA10, OmniVision Technologies Inc., Carolina, America) with a square cross-section of 0.65*0.65mm, and two Led lights with a cross-section of 0.35*0.65mm. The high-stiffness proximal section of the bronchoscope catheter is composed of the braided mesh structure made of stainless-steel material and outer thin thermoplastic urethanes (TPU) layer. The low-stiffness distal section is composed of the snake tube made of laser-cut stainless-steel material and outer TPU layer.

revised content:

Supplementary Note 1.

Supplementary Fig. 1 offers the CAD model of our AI co-pilot bronchoscope robot,

designed to incorporate the plug-and-play bronchoscope catheters. The catheter is fixed on the guide seat, driven by two pairs of antagonistic tendons to achieve omnidirectional bending deformation of the distal section. The tendons are divided into four directions along the grooves of the guide seat and connect with the upper magnet holders. The steering control system consists of four linear motors (LA50-021D, Inspire-Robots, Beijing, China) for pulling the tendons and four force sensors (QLA414, FUTEK, California, America) for measuring the driving force. One side of each force sensor is connected to the linear motor through the motor flange, and another side is connected to the lower magnetic holder. Two sets of magnets are respectively installed inside the upper and lower magnet holders. Based on the magnetic adsorption force, the bronchoscope catheter can be quickly installed on the steering control system. The linear motors are installed on the motor fixture which connects with the electric slide (EZSM3E040AZMK, Oriental Motor, Tokyo, Japan) by the mounting shell to achieve the feed movement of the bronchoscope catheter. The electric slide is actuated by the slide driver (AZD-KD, Oriental Motor, Tokyo, Japan), and installed on the robotic arm (UR5, Universal Robots, Odense, Denmark) to achieve the large range of pose adjustment of the bronchoscope robot.

Supplementary Fig. 1. The CAD model of the AI co-pilot bronchoscope robot system. The system consists of a robotic arm, an electric slide, a steering control system, and plug-and-play bronchoscope catheters. The steering control system is composed of four linear motors to steer the catheter, and four force sensors to measure the actuation force. The steering control system and bronchoscope catheter are connected by magnetic adsorption force for rapid catheter replacement. The electric slide is used to feed the bronchoscope catheter, while the UR robotic arm is used to achieve the large range of pose adjustment of the bronchoscope robot.

The bronchoscope catheter consists of the high-stiffness proximal section and the low-stiffness distal section. The proximal section uses the braided mesh structure for increased stiffness, while the distal section uses the snake tube made of stainless steel for steering control. The catheter is covered with the thin thermoplastic urethanes (TPU) layer for waterproofing. To improve the application range of the bronchoscope robot, a 3.3mm catheter with a 1.2mm working channel and a 2.1mm catheter are designed, enabling access to the ninth deeper generation bronchi for average adult patients. The two catheters are both installed with the micro camera (OCHTA10, OmniVision Technologies Inc., Carolina, America) with a square cross-section of 0.65*0.65mm, and two Led lights with a cross-section of 0.35*0.65mm. The proximal length of the two catheters is 650mm, while the snake bone length of the 3.3mm and 2.1mm catheter are respectively 30mm and 25mm. The distal section of the two catheters can achieve an omnidirectional bending of about 180 degrees for deep lung examination.

Comment 10:

Further, the authors claim that this robot will address disparities in healthcare; however, the proposed robot does not seem to make progress toward accessibility of bronchoscopy procedures in underdeveloped regions based on the types of materials.

Response to Comment 10:

To demonstrate the potential of the robot used in underdeveloped regions, we counted the robotic system's composition and calculated the total cost as depicted in **Supplementary Table 4** in the revised manuscript. The minimum total cost of our system is \$4168, which can be afforded by most healthcare institutions, thereby making our technology accessible on a broader scale. Even we retrofit four advanced high-precision force sensors in our system to monitor actuation force during the bronchoscopy, the total cost of **\$13476** still remains low compared with Monarch Platform (**\$500,000**) and Ion Endoluminal System (**\$204,000 per year**).

Additionally, we present an alternative force sensor option, the LM15 sensor from Shenzhen Lichi Sensing Technology Co., LTD, with a capacity of 20N and a resolution of 0.1N. Priced at **\$137** per unit, this alternative retains the crucial functionality of force sensing while significantly reducing the overall cost. The cumulative effect of this substitution brings the total cost of the robotic system to approximately **\$4716**.

revised content:

Supplementary Table 4. Robot costing. The standard version of the system without force sensors has the cost of \$4168, while the advanced version of the system with force sensors has the cost of \$13476.

Equipment or mechanical parts	Model	unit price	number	total price	remarks
linear motor	LA-50	\$273	4	\$1,092	used for steering the catheter
electric slide	EZSM3E040	\$861	1	\$861	used for feeding the catheter
	AZMK				
slide driver	AZD-KD	\$350	1	\$350	used for driving the electric slide
micro camera	OCHTA10	\$525	1	\$525	used for endoscopic imaging
snake tube	—	\$410	1	\$410	used to make the distal section
braided mesh tube	—	\$410	1	\$410	used to make the proximal section
waterproof rubber	—	\$110	1	\$110	used as a waterproof layer
machined parts	—	\$410	1	\$410	used as connecting parts or shells
force sensor*	QLA414	\$2,327	4	\$9,308	used for force limit
Standard version price		\$4168		Advanced version price	\$13476

"*" means optional, the device can be eliminated or substituted.

Comment 11:

How do the dynamics of the bronchoscope robot play a role in controlling and keeping the bronchoscope centered?

Response to Comment 11:

The steering control of the bronchoscope robot is mainly dependent on the proposed kinematic model, which ignores the dynamic characteristics of the robot. The kinematic model maps the configuration parameters of the catheter to the actuation state. Based on the model, we proposed the control method to steer the bronchoscope catheter in the lung, This content is detailedly depicted in **Supplementary Note 3**.

Comment 12:

In some of the movies, there seems to be overshooting in the AI system, will this be safe? An evaluation of the overshooting effect would clarify this.

Response to Comment 12:

We appreciate the reviewer's insightful question. In our movies of in-vitro and in-vivo experiments, there seems to be a periodic oscillation of our robot from one side of bronchial inner wall to the other side, just like overshooting of our AI system. Actually, this situation can be seen as a dynamic tracking process caused by the respiratory movement of lung, rather than overshooting of our AI system. When bronchoscopy is performed, due to the interaction of respiratory movement and the movement of the robot, the tracking target of the robot's head is constantly changing, which requires the robot to track the target in real time. The overshooting shown in the video is mainly the error when tracking dynamic targets. In order to analyze the overshooting and its

influence on safety, we have supplemented an AI-assisted self-centering control experiment as depicted in **Fig. R17a**. The robot catheter is controlled in advance to one side of the bronchial inner wall. Under AI control, the robot automatically bends towards the lumen center, and in this process, we analyze the robot's motion trajectory by recording image error. As can be seen in **Fig. R17b**, our robot can turn back to center within 2s and hardly ever overshoots. Besides, as can be seen from the in-vitro and in-vivo bronchoscopy experimental results, compared with teleoperation by expert, AI control still has better tracking accuracy, which proves its safety. In addition, we have also added some sentences to discuss the self-centering control performance of our AI system in the revised manuscript, please check **Supplementary Note 12** for details.

Fig. R17 (Supplementary Fig. 23) AI-assisted self-centering control experiment.
a, Four sides of bronchial inner wall where the robot catheter is controlled in advance to.
b, Image error curves in Up-Down and Left-Right directions.

Comment 13:

Furthermore, does the use of only five discrete commands limit the motion of the robot to two planes or can the robot be bent with multiple tendons being pulled at a time? A clarification of the control strategy would clarify this.

Response to Comment 13:

We appreciate the reviewer's kind suggestion. The use of only five discrete commands does not limit the motion of the robot to two planes, and the robot can be bent with multiple tendons being pulled at a time. As shown in **Fig. R18 (Supplementary Fig.**

24), in a process of bronchoscopy, when the discrete human command is constantly “right”, the steering actions predicted by the policy network are not limited in horizontal plane but can be any direction according to the position of bronchial lumen in bronchoscopic image and the human command. For example, at $t=0s$, according to the human command “right”, the policy network identifies doctor’s intention of truing right and outputs a steering action that points to the center of right bronchial lumen. At $t=0.45s$, the policy network keeps steering the robot to the right bronchial lumen. At $t=1.05s$, the robot enters into the right lumen and the policy network outputs a slightly left steering action for collision avoidance, although the human command is still “right”, leading to a safe bronchoscopy procedure. As a result, although the human commands are discrete, the steering actions predicted by the policy network are continuous and not limited to two planes, which are finally converted to continuous and safe tendon displacements for robot control.

Fig. R18 Visualizaiton of AI-human shared control strategy during bronchoscopy.

In addition, the above contents have been added in **Supplementary Note 12**, please check it for details.

Comment 14:

Is the robot arm that holds the bronchoscope robot also controlled with the AI co-pilot and human teleoperator or is the robot arm set in a fixed position throughout the whole procedure? Could a different robot arm position result in a larger image error in the experiments?

Response to Comment 14:

In this work, the robot arm is set in a fixed position throughout the whole procedure, which can be replaced by a fixed trestle. Primarily, the inlet direction of the bronchoscope catheter should conform to the human body (the valve) as much as possible. To this end, during the in-vivo experiments, we adjust the robotic arm to let the bronchoscope catheter towards the valve of the pig to facilitate the subsequent catheter feed movement. Conversely, the improper adjustment of the robot arm's pose—specifically, the notable misalignment between the catheter's inlet direction and the porcine valve, can impact the feeding movement (unsmooth) of the catheter and even increase image errors.

Comment 15:

How does the AI co-pilot affect the overall speed of the procedure?

Response to Comment 15:

We apologize for not providing detailed discussion on the overall efficiency of our AI co-pilot system. In the in-vitro and in-vivo experiments, we set the forward speed of the robot to be constant at 2 mm/s. For both AI co-pilot and teleoperation bronchoscopy, the start/stop of the robot can be manually controlled by the teleoperator and the forward speed can be manually adjusted in the software. However, in practical use, the speed of bronchoscopy is influenced by factors such as the doctor's experience, the difficulty of robot operation, and the patient's current respiratory state. Under teleoperation control, the operator needs to continuously adjust the bronchoscope's posture based on the environmental conditions, which may involve temporarily stopping moving forward to adjust the robot's posture. Under AI shared control, the adjustment of the robot's posture is dynamically ensured by the AI algorithm, and the doctor only needs to select the next airway branch, without the need to continuously monitor the robot's posture. This can improve the efficiency of the doctor's operation to some extent.

Fig. R19 Learning process of the novice doctors.

Specifically, according to the results of in-vitro experiment in **Fig. R19 (Fig. 4c)**, if a novice performs teleoperation, the speed of bronchoscopy operation will be significantly slower than that of an expert doctor, and the quality of bronchoscopy will be inferior to that of an expert. However, As depicted in **Fig. R19 (Fig. 4c)** and **Fig. R20 (Fig. 5e)**, when a novice uses AI assistance, after alleviating a substantial cognitive

load, the speed can approach or even outperform that of an expert while ensuring better bronchoscopy quality (lower image error) than an expert doctor.

Fig. R20 Image error during bronchoscopy in live porcine lung.

Comment 16:

In the in-vivo experiments, the number of interventions is reported as a result which characterizes autonomy. However, it is stated that the number of interventions is lower than those interventions of the expert. This statement seems counterintuitive since there is constant human intervention in the case of the expert surgeon. A definition of human intervention by the authors would be helpful here.

Response to Comment 16:

Thanks to the reviewer’s insightful suggestion. We apologize for not providing clear definition and explanation of human intervention. In our manuscript, we define human intervention as the number of switching actions of doctor’s hand. Specifically, for teleoperation mode without AI assistance, we record the number of time stamps where the tendon displacements of linear motors change from last time stamp as the number of human interventions. For AI shared control mode, we record the number of time stamps where discrete human command (e.g. up, down, left, right or forward) changes from last time stamp as the number of human interventions. The human intervention ratio (HIR) is defined as the ratio of the number of human interventions T_{interv} to the total number of time stamps T_{total} recorded in the whole bronchoscopy procedure:

$$HIR = T_{interv}/T_{total}$$

As shown in **Fig. R21** (**Fig. 5g** in the maintext), the HIR under AI shared control by novice doctor is significantly lower than that under teleoperation control by expert doctor. It demonstrates that AI-assisted bronchoscopy can effectively minimize the physical strain and cognitive burden on doctors. Please check **Supplementary Note 9** for details.

Fig. R21 Statistical results of the human intervention ratios, determined by the number of time stamps where the action of doctor’s hand changes from last time stamp relative to the total time stamps recorded in the whole bronchoscopy procedure.

Reviewer #3 (Remarks to the Author):

General Comment:

Thank you for giving me the opportunity to review this very interesting manuscript.

Response to General Comment:

We are grateful for your careful review and valuable suggestions, which have enabled us to greatly improve our article. We are pleased that the referee finds our work is interesting. We have carefully considered all the comments and suggestions and have made corresponding improvements as presented in this response letter. Thank you again.

Comment 1:

In Introduction: please, improve the state of the art by citing the most relevant articles on Monarch Platform and Ion Endoluminal System.

Response to Comment 1:

Thanks for the thorough comment of reviewers, we have added some references [17-21] in the revised manuscript as the supplement to the Monarch Platform and Ion Endoluminal System.

revised content:

Introduction (underlined text indicates added content)

The Monarch Platform is equipped with an internal bronchoscope catheter with 4.2 mm diameter, and an external sheath of 6 mm. Its subtle steering control and flexibility, allowing for deeper access into the peripheral regions of the lungs, surpassing conventional bronchoscopes¹⁷ (9th vs. 6th airway generations). In parallel, the Ion Endoluminal System boasts a fully articulated 3.5mm catheter with a 2mm working channel, enhanced stability, superior flexibility, and the added advantage of shape perception¹⁸. Notably, studies indicate that these platforms exhibit a favorable diagnostic yield, ranging from 81.7% to 92%, for lung nodules with sizes between 14.8 mm and 21.9 mm¹⁹⁻²¹. Moreover, the complication rates reported are minimal. These findings suggest that these platforms play a transformative role in the future management of pulmonary conditions. In addition to the Monarch and Ion platforms, several other bronchoscope robotic systems developed by academic institutions are under development or in early-stage research to address sensing and control issues for doctors²²⁻²⁷. Despite its advantages, current telerobotic bronchoscopy faces several challenges, including the learning curve and lack of autonomy.

[17] Murgu, S. D. Robotic assisted-bronchoscopy: technical tips and lessons learned from the initial experience with sampling peripheral lung lesions. *BMC Pulm. Med.* **19**, 1-8 (2019).

[18] Simoff, M. J. et al. Shape-sensing robotic-assisted bronchoscopy for pulmonary nodules: initial multicenter experience using the Ion™ Endoluminal System. *BMC Pulm. Med.* **21**, 1-13 (2021).

[19] Benn, B. S., Romero, A. O., Lum, M. & Krishna, G. Robotic-assisted navigation bronchoscopy as a paradigm shift in peripheral lung access. *Lung* **199**, 177-186 (2021).

[20] Kalchier-Dekel, O. et al. Shape-sensing robotic-assisted bronchoscopy in the diagnosis of pulmonary parenchymal lesions. *Chest* **161**, 572-582 (2022).

[21] Ost, D. et al. Prospective multicenter analysis of shape-sensing robotic-assisted bronchoscopy for the biopsy of pulmonary nodules: results from the PRECISE study. *Chest* **160**, A2531-A2533 (2021).

Comment 2:

On line 125: please add diameter of Monarch Platform and Ion Endoluminal System.

Response to Comment 2:

Thanks for the thorough comments of reviewers. The Ion Endoluminal system, has an outer diameter of 3.5mm and a working channel of 2mm, but the camera probe needs to be removed in advance to insert the working channel. The Monarch Platform features an internal bronchoscope catheter (4.2 mm) and an external sheath (6 mm), as well as a more flexible and subtle steering control that can reach farther into the periphery of the lungs than conventional bronchoscope (9th vs. 6th airway generations). We have added the diameter of Monarch Platform and Ion Endoluminal System in the revised manuscript.

revised content:

Introduction (underlined text indicates added content)

The development of robotic platforms and devices for bronchoscopy has seen significant progress in recent years, with systems such as the Monarch Platform¹⁵ and Ion Endoluminal System¹⁶ leading the way. The Monarch Platform is equipped with an internal bronchoscope catheter with 4.2 mm diameter, and an external sheath of 6 mm. Its subtle steering control and flexibility, allowing for deeper access into the peripheral regions of the lungs, surpassing conventional bronchoscopes¹⁷ (9th vs. 6th airway generations). In parallel, the Ion Endoluminal System boasts a fully articulated 3.5mm catheter with a 2mm working channel, enhanced stability, superior flexibility, and the added advantage of shape perception¹⁸.

[17] Murgu, S. D. Robotic assisted-bronchoscopy: technical tips and lessons learned from the initial experience with sampling peripheral lung lesions. *BMC Pulm. Med.* **19**, 1-8 (2019).

[18] Simoff, M. J. et al. Shape-sensing robotic-assisted bronchoscopy for pulmonary nodules: initial multicenter experience using the Ion™ Endoluminal System. *BMC Pulm. Med.* **21**, 1-13 (2021).

Comment 3:

How many novice doctors tested your AI robot? Who are the novice users? Do they have any bronchoscopy experience?

Response to Comment 3:

Thanks for the reviewer's thorough comment. There are two novice doctors participating in the experiments, including a medical intern and an attending doctor from the School of Medicine, Zhejiang University, Hangzhou, China. The results from the intern are depicted in **Fig. 4c**, while the results from the attending physician are showcased in **Fig. 4c, d, e**, as well as **Fig. 5**.

The novice user is defined as one kind of doctors who lack significant experience in performing robotic bronchoscopy operations, and have watched operational videos of the bronchoscope robot. The operation experience of participants in the experiments is depicted in **Supplementary Table 3**.

revised content:

Supplementary Table 3. The operation experience of participants in the experiments. The medical intern (novice 1), the attending doctor (novice 2) and the expert doctor (expert) both come from the School of Medicine at Zhejiang University, Hangzhou, China.

Participant	Professional title	Entire period of operation	Manual operations	Robot teleoperations
novice 1	medical intern	no	no	2 demonstrations
novice 2	attending doctor	<5 years	<100 cases per year	2 demonstrations
expert	chief doctor	>20 years	>200 cases per year	>100 trials

Comment 4:

The Discussion is too short. It should be improved. Please, add some context from the evidence of published literature on Monarch Platform and Ion Endoluminal System.

Response to Comment 4:

Follow your kind suggestions, we have expanded the **Discussion** referred to some published literatures on Monarch Platform and Ion Endoluminal System.

revised content:**Discussion and Outlook (the first paragraph)**

Bronchoscopic intervention is preferred for sampling suspected pulmonary lesions owing to its lower complications. In recent times, robot-assisted technologies have been introduced into the bronchoscopic procedures to enhance the maneuverability and stability during lesion sampling, such as the Monarch Platform and the Ion Endoluminal System. However, due to the high cost of robotic bronchoscope systems and the expertise required by doctors, the proliferation of this technology in underdeveloped

regions is limited. Our study presents a comprehensive AI co-pilot bronchoscope robot designed to improve the safety, accuracy, and efficiency of bronchoscopy procedures. The proposed system, integrating a shared control algorithm and state-of-the-art domain adaptation and randomization approaches, bridges the gap between simulated and real environments, ensuring generalizability across various clinical settings. Moreover, this co-pilot bronchoscope robot enables novice doctors to perform bronchoscopy as competently and safely as experienced specialists, reducing the learning curve and ensuring consistent quality of care.

Comment 5:

I suggest adding some paragraphs to frame your work in terms of explainable AI. What about the limitations of your work?

Response to Comment 5:

We appreciate the reviewer's insightful suggestions. To analyze the explainable reason for decision-making of our AI co-pilot system during bronchoscopy procedures, we have conducted an experiment of the interpretability of the proposed policy network using three styles of image pairs. We generate gradient-weighted class activation maps (GradCAM) from the last convolutional layer of the policy network as the network's attention and visualized the fusion image by overlaying GradCAM onto the original images. Highlight regions mean the key clues that our policy network put attention to for making decision. As shown in **Fig. R22 (Supplementary Fig. 27)** it's observed that our network has learned to focus on bronchial lumens, and as the distance between the robot and bronchial wall increases, the attention value becomes larger. This indicates that our network concentrates on the structural information of airway and utilizes it to predict safe actions, keeping the center of bronchial lumen at the center of the image. As a result, it's reasonable that the AI co-pilot robot is able to remain centered in the airway and stay as far as possible away from bronchial wall during bronchoscopy procedures. These contents have been added in **Discussion and Outlook**, please check it for details.

Fig. R22 (Supplementary Fig. 27) Gradient-weighted Class activation maps (GradCAM) from the last convolutional layer of our policy network. Three different styles of bronchoscopic images are chosen for validation.

The limitations of our work can be listed as follows: Our bronchoscope robot relies upon tendon actuation for precise steering control, feeding into the deep lungs by the "follow-the-tip" motion. In alignment with this methodology, the proximal section of the catheter is engineered to exhibit a substantially increased stiffness in comparison to the distal section. However, large bending angle (approaching 180 degrees) of the distal section presents great challenges in effecting smooth feed movement of the catheter, particularly when negotiating the upper pulmonary regions. In addition, it is essential to assess the robustness of the proposed method in a broader range of clinical scenarios, including patients with varying bronchial anatomies, pathologies, or respiratory conditions. Extensive testing on a diverse patient population will be necessary to validate the applicability of the intelligent bronchoscope robot in real-world settings. The integration of additional sensing modalities can be further deployed, such as ultrasound or optical coherence tomography, could provide complementary information to guide the bronchoscope robot. Fusing multiple data sources could improve the accuracy and safety of AI-assisted steering, offering more comprehensive diagnostic and therapeutic support.

revised content:

Discussion and Outlook (the third and fourth paragraphs)

Despite these promising results, there are several areas for future research and development. Our bronchoscope robot relies upon tendon actuation for precise steering control, feeding into the deep lungs by the "follow-the-tip" motion⁴¹⁻⁴². In alignment with this methodology, the proximal section of the catheter is engineered to exhibit a substantially increased stiffness in comparison to the distal section. However, large bending angle (approaching 180 degrees) of the distal section presents great challenges in effecting smooth feed movement of the catheter, particularly when negotiating the upper pulmonary regions. Soft untethered magnetic catheter has the potential to improve the ability of bronchoscope for deep lung examination, and is worthy of study. In addition, it is essential to assess the robustness of the proposed method in a broader range of clinical scenarios, including patients with varying bronchial anatomies, pathologies, or respiratory conditions. Extensive testing on a diverse patient population will be necessary to validate the applicability of the intelligent bronchoscope robot in real-world settings. The integration of additional sensing modalities can be further deployed, such as ultrasound or optical coherence tomography, could provide complementary information to guide the bronchoscope robot. Fusing multiple data sources could improve the accuracy and safety of AI-assisted steering, offering more comprehensive diagnostic and therapeutic support.

Besides, the explainability of our AI co-pilot system is discussed by analyzing the reason for decision-making of AI during bronchoscopy procedures. We conduct an experiment of the interpretability of the proposed policy network using three styles of image pairs. We generate gradient-weighted class activation maps (GradCAM) from the last convolutional layer of the policy network as the network's attention and visualized the fusion image by overlaying GradCAM onto the original images. Highlight regions mean the key clues that our policy network put attention to for making decision. As shown in Supplementary Note 13, it's observed that our network has learned to focus on bronchial lumens, and as the distance between the robot and bronchial wall increases, the attention value becomes larger. This indicates that our network concentrates on the structural information of airway and utilizes it to predict safe actions, keeping the center of bronchial lumen at the center of the image. As a result, it's reasonable that the AI co-pilot robot is able to remain centered in the airway and stay as far as possible away from bronchial wall during bronchoscopy procedures.

Comment 6:

In Supplementary Note 4. I suggest adding details on how the segmentation was performed.

Response to Comment 6:

We apologize for not providing detailed information on how the airway segmentation is performed. In the process of establishing virtual bronchoscopy environment, the airway segmentation from pre-operative CT volume is a key step for simulating

bronchus and extracting centerlines as reference paths. In this study, we utilize the region-growing algorithm to segment airway tree from CT volume, by manually placing a seed within the trachea. Then the adjacent regions can be automatically annotated as the same label if the Hounsfield Unit (HU) values are similar and the segmentation can be completed. In practice, region-growing algorithm can be easily implemented by the AirwaySegmentation Module of 3D Slicer software.

Specifically, in **Fig. R23**, a thorax CT volume is loaded in 3D Slicer software at first. Then a 3D fiducial (or seed), shown as AirwayFiducial-1, should be manually added and placed in the target region to segment, i.e. airway region in this study. After that, the region-growing algorithm is triggered by pressing the “Apply” button. When the segmentation is completed, a 3D mask of airway tree can be generated, as visualized in the window. Lastly, the 3D airway mask is processed by several morphological operations to get a bronchus shell model, which is used to establish virtual bronchoscopy environment for training our policy network. The above content is added to **Supplementary Note 4**.

Fig. R23 AirwaySegmentation module of 3D Slicer and an example of airway segmentation from a thorax CT volume.

REVIEWERS' COMMENTS

Reviewer #1 (Remarks to the Author):

Thank you for the opportunity to review this revised manuscript from Zhang and Liu et al. The authors have a number of changes based on reviewer feedback which have significantly improved the manuscript.

I had a few remaining minor points:

In the section regarding in vitro experiments, lines 206-210 mention experience levels of the various operators. It might be helpful to include a bit more clarification in the text here. For instance, rather than say the attending doctor has "a little experience" maybe say less than 50 cases compared to the chief doctor who has >200 cases or something like that.

In addition, it is worth noting that the steering mechanism of this platform (seems like some sort of joystick) is considerably different that control of a standard, hand-held bronchoscopy (or even current robotic bronchoscopy systems). Therefore it is hard to know whether previous bronchoscopy experience is really relevant to these experiments. It might be worth mentioning this in the limitations section.

Reviewer #2 (Remarks to the Author):

The authors have addressed my comments.

Reviewer #3 (Remarks to the Author):

Thank you for giving me the chance to review the revised version.

The authors developed a robotic AI-assisted platform for bronchoscopy which is smaller than the commercially available solution by Auris Robotics (Monarch) and Intuitive Surgery (Ion System).

During in-vitro tests in a simulation environment, the results have shown that the proposed solution achieved a higher score than state-of-the-art.

During in vivo tests the physician performed better with the AI-assisted mode enabled than the expert participant.

There are some typos, e.g.:

Row 101-102, please check.

Row 131: inverse kinematics, not inverse kinetics

Row 165: definations

Rows 296-302; 319 -322: please check and refine English

I suggest authors to check and improve English in the revised version

Response to Reviewers' Comments on "NCOMMS-23-26020-A"

We would like to thank the editors for their constructive feedback, which have significantly contributed to the improvement of our manuscript. We have thoroughly considered and addressed each comment from both the editor and reviewers. Point-by-point responses to their comments are provided below, with corresponding changes highlighted in the revised manuscript in red for easy tracking

Reviewer #1

Thank you for the opportunity to review this revised manuscript from Zhang and Liu et al. The authors have a number of changes based on reviewer feedback which have significantly improved the manuscript.

Response to Comment:

We are grateful to the referee for their thorough review of our manuscript and their valuable comments. We have made substantial revisions to our manuscript. Please find our detailed responses below.

General comments:

Comment 1:

I had a few remaining minor points:

In the section regarding in vitro experiments, lines 206-210 mention experience levels of the various operators. It might be helpful to include a bit more clarification in the text here. For instance, rather than say the attending doctor has "a little experience" maybe say less than 50 cases compared to the chief doctor who has >200 cases or something like that.

Response to Comment 1:

We appreciate the reviewer for the kind suggestion. We have revised the content in the revised manuscript to include further clarification regarding the operators. Please check Page 8 Paragraph 2 for more details.

Revised content, on page 8 of the main text:

Simulation Results and in-vitro Evaluation (the last paragraph)

To assess the proposed AI co-pilot bronchoscope robot, experiments were conducted on a bronchial phantom made of silica gel replicating structured derived from human CT lung data (Fig. 4a). A crank-rocker mechanism-based breathing simulation system (Supplementary Note 7, Supplementary Fig. 11) was designed to emulate human respiration (15 cycles per minute). Two bronchial phantoms with distinct bronchial structures were employed for in vitro evaluation (Fig. 4b). An expert (chief doctor) and a novice doctor (medical intern) were invited to perform bronchoscopic procedures using the robot without the AI co-pilot as a benchmark, while another novice doctor

(attending doctor) also participated using the robot with the AI co-pilot. All procedures were performed on the same path using the teleoperator. The medical intern had no experience with bronchoscopy, while the attending doctor had a little experience (<5 years and <100 cases per year, compared to the chief doctor, who had >20 years of experience and >200 cases per year), as indicated in Supplementary Table 1. They were both presented with two demonstrations of robotic intubation, with and without the AI co-pilot, to learn how to operate the system.

Comment 2:

In addition, it is worth noting that the steering mechanism of this platform (seems like some sort of joystick) is considerably different that control of a standard, hand-held bronchoscopy (or even current robotic bronchoscopy systems). Therefore it is hard to know whether previous bronchoscopy experience is really relevant to these experiments. It might be worth mentioning this in the limitations section.

Response to Comment 2:

We appreciate the reviewer's for this insightful comment. In fact, the doctor's previous bronchoscopy experience not only involves familiarity with the steering mechanism of existing bronchoscope controller (e.g. hand-held or current robotic bronchoscopy), but also includes a high-level understanding of bronchoscopic images (e.g. real-time assessment of potential risks and determination of safe adjustments). During bronchoscopy, the bronchoscope should be inserted and manipulated gently to avoid abrupt or forceful movements, which can cause discomfort or injury to airway structures. Additionally, maintaining a central position during bronchoscopy allows better visualisation of the airway anatomy and helps prevent injury to the airway mucosa or other structures. Generally, expert doctors have a better high-level understanding than novice doctors owing to their superior clinical experience. On this basis, the relationship between previous bronchoscopy experience and our experiments can be analyzed in two aspects. On one hand, if we assume that the familiarity with current steering mechanism is not relevant to the proficiency in using our system (i.e. neither expert nor novice is familiar with our system), our experimental results demonstrate that the novice doctor with AI co-pilot achieves comparable control performance to the expert, highlighting that AI can bridge the gap in high-level understanding between novices and experts. On the other hand, if we assume that familiarity with current steering mechanism is positively relevant to the proficiency in using our system (i.e. experts are more familiar with our system than novices), our experimental results can also support the same conclusions.

In this study, our approach uses TOUCH as the teleoperator to input human commands, which differs from the traditional rocker used in hand-held bronchoscopes and the gamepad used in some robotic systems. According to the above analysis and the reviewer's insightful suggestion, it's worth studying the relevance between the familiarity with current steering mechanism and the proficiency in using our system,

because it helps us to know the learning cost of our steering mechanism and find the potential of our system for better assisting novice doctors to reduce the learning curve. Thus, in the future, further research is warranted to investigate the relationship between the previous experience of doctors in current teleoperators and the proficiency of operating our system. We have incorporated this point into the Discussion section of the revised manuscript.

Revised content, on page 10 of the main text:

Discussion and Outlook

Despite these promising results, there are several areas for future research and development. Our bronchoscope robot relies upon tendon actuation for precise steering control and is fed into the deep lungs by means of "follow-the-tip" motion. In alignment with this methodology, the proximal section of the catheter is engineered to exhibit a substantially increased stiffness in comparison to the distal section. However, a large bending angle (approaching 180 degrees) of the distal section presents great challenges in effecting smooth feed movement of the catheter, particularly when negotiating the upper pulmonary regions. A soft untethered magnetic catheter design has the potential to improve the capabilities of bronchoscopy for deep lung examination and is worthy of study. In addition, it is essential to assess the robustness of the proposed method in a broader range of clinical scenarios, including patients with varying bronchial anatomies, pathologies, or respiratory conditions. Extensive testing on a diverse patient population will be necessary to validate the applicability of the intelligent bronchoscope robot in real-world settings. Considering the difference in teleoperators between our AI co-pilot system and existing robotic or hand-held bronchoscopy systems, the relevance between the previous experience of doctors in current teleoperators and the proficiency of operating our system is worth further studying. The integration of additional sensing modalities, such as ultrasound or optical coherence tomography, can also be considered to provide complementary information to guide the bronchoscope robot. Fusing multiple data sources could improve the accuracy and safety of AI-assisted steering, offering more comprehensive diagnostic and therapeutic support.

Reviewer #2 (Remarks to the Author):

General Comment:

The authors have addressed my comments.

Response to General Comment:

We greatly appreciate the referee's comprehensive review and recognition of our work.

Reviewer #3 (Remarks to the Author):

General Comment:

Thank you for giving me the chance to review the revised version.

The authors developed a robotic AI-assisted platform for bronchoscopy which is smaller than the commercially available solution by Auris Robotics (Monarch) and Intuitive Surgery (Ion System).

During in-vitro tests in a simulation environment, the results have shown that the proposed solution achieved a higher score than state-of-the-art.

During in vivo tests the physician performed better with the AI-assisted mode enabled than the expert participant.

Response to General Comment:

We are grateful for the referee's careful review and valuable suggestions, which have enabled us to greatly improve our article. We have carefully revised all the comments and suggestions as presented in this response letter.

Comment 1:

There are some typos, e.g.:

Row 101-102, please check.

Row 131: inverse kinematics, not inverse kinetics

Row 165: definations

Rows 296-302; 319 -322: please check and refine English

Response to Comment 1:

Thanks for the thorough comments of the reviewer. We have carefully checked the paper and revised all the typos in the revised manuscript. Please check it in the revised manuscript.

Revised content:

For Row 101-102, we have checked and refined the sentence "..., it is anticipated that the cost and logistical barriers associated with the adoption of these platforms will decrease, confronting the challenge of medical resource disparities and contributing to the improvement of global health outcomes." as "It is anticipated that the cost and logistical barriers associated with the adoption of such platforms will decrease in the future, helping to overcome the challenge of medical resource disparities and contributing to the improvement of global health outcomes. (Row 97-100)"

For Row 131, we have replaced the typo "inverse kinetics" by "inverse kinematics (Row 128)".

For Row 165, we have revised the typo "definations" to "definitions (Row 164)".

For Row 296-302, we have checked and refined the sentences “Besides, the explainability of our AI co-pilot system is discussed by analyzing the reason for decision-making of AI during bronchoscopy procedures. We conduct an experiment of the interpretability of the proposed policy network using three styles of image pairs. We generate gradient-weighted class activation maps (GradCAM) from the last convolutional layer of the policy network as the network's attention and visualized the fusion image by overlaying GradCAM onto the original images. Highlight regions mean the key clues that our policy network put attention to for making decision.” as “In addition, the explainability of our AI-copiloted system was investigated by analysing the reasons for the decision-making of the AI during bronchoscopic procedures. We conducted an experiment on the interpretability of the proposed policy network using three styles of image pairs. We generated gradient-weighted class activation maps (GradCAM) from the last convolutional layer of the policy network to represent the network's attention and visualised the fused images by overlaying the GradCAM results onto the original images. In the resulting images, highlighted regions indicated the key clues that our policy network paid attention to when making decisions. (Row 315-322)”

For Row 319-322, we have checked and refined the sentences “In hardware level, the bronchoscope robot employs tendon-driven mechanics, leveraging four linear motors to precisely steer the bronchoscope catheter, and an electric slide for feed movement. Additionally, our robot system boasts an innovative magnetic adsorption method for rapidly replacement of the catheter.” as “At the hardware level, the bronchoscope robot employs tendon-driven mechanics, leveraging four linear motors to precisely steer the bronchoscope catheter and an electric slide for feed movement. Additionally, our robotic system boasts an innovative magnetic adsorption method for rapid replacement of the catheter. (Row 339-342)”

Comment 2:

I suggest authors to check and improve English in the revised version.

Thank you for giving me the opportunity to review this very interesting manuscript.

Response to Comment 2:

Thanks for the kind suggestion of the reviewer. We have improved English Language using Nature Research Editing Service in the revised manuscript. The verification code is **FA72-CDB1-8B8E-490E-E0F3** which can be verified on the SNAS website. The following is the certificate.

This document certifies that the manuscript
AI Co-pilot Bronchoscope Robot

prepared by the authors
Jingyu Zhang

was edited for proper English language, grammar, punctuation, spelling, and overall style
by one or more of the highly qualified native English speaking editors at SNAS.

This certificate was issued on **November 7, 2023** and may be verified
on the SNAS website using the verification code **FA72-CDB1-8B8E-490E-E0F3**.

Neither the research content nor the authors' intentions were altered in any way during the editing process. Documents receiving this certification should be English-ready for publication; however, the author has the ability to accept or reject our suggestions and changes. To verify the final SNAS edited version, please visit our verification page at secure.authorservices.springernature.com/certificate/verify. If you have any questions or concerns about this edited document, please contact SNAS at support@as.springernature.com.